# Acute control of the sleep switch in *Drosophila* reveals a role for gap junctions in regulating behavioral responsiveness

**Michael Troup[1†], Melvyn HW Yap[1†], Chelsie Rohrscheib[1], Martyna J Grabowska[1], Deniz Ertekin[1], Roshini Randeniya[1], Benjamin Kottler[1,2], Aoife Larkin[1,3], Kelly Munro[1], Paul J Shaw[4], Bruno van Swinderen[1]\***

[1]Queensland Brain Institute, The University of Queensland, Brisbane, Australia; [2]King's College London, London, United Kingdom; [3]University of Cambridge, Cambridge, United Kingdom; [4]Washington University School of Medicine, St Louis, United States

**Abstract** Sleep is a dynamic process in most animals, involving distinct stages that probably perform multiple functions for the brain. Before sleep functions can be initiated, it is likely that behavioral responsiveness to the outside world needs to be reduced, even while the animal is still awake. Recent work in *Drosophila* has uncovered a sleep switch in the dorsal fan-shaped body (dFB) of the fly's central brain, but it is not known whether these sleep-promoting neurons also govern the acute need to ignore salient stimuli in the environment during sleep transitions. We found that optogenetic activation of the sleep switch suppressed behavioral responsiveness to mechanical stimuli, even in awake flies, indicating a broader role for these neurons in regulating arousal. The dFB-mediated suppression mechanism and its associated neural correlates requires *innexin6* expression, suggesting that the acute need to reduce sensory perception when flies fall asleep is mediated in part by electrical synapses.
DOI: https://doi.org/10.7554/eLife.37105.001

**\*For correspondence:**
b.vanswinderen@uq.edu.au

[†]These authors contributed equally to this work

**Competing interests:** The authors declare that no competing interests exist.

## Introduction

Most animals sleep, and recent research suggests that some proposed sleep functions may be deeply conserved across all animals (*Cirelli, 2009*; *Kirszenblat and van Swinderen, 2015*; *Zimmerman et al., 2008*). These functions range from cellular stress regulation in nematodes (*Hill et al., 2014*; *Nelson et al., 2014*) to synaptic homeostasis and memory consolidation in mammals (*Rasch and Born, 2013*; *Tononi and Cirelli, 2006*; *Tononi and Cirelli, 2014*). Other proposed functions of sleep include maintaining cellular health, by clearing of protein debris for example (*Xie et al., 2013*). These conserved cellular processes might have been integrated through evolution into distinct stages of the mammalian sleep cycle, to accommodate multiple more recent sleep functions, such as synaptic scaling and memory consolidation, simultaneously (*Kirszenblat and van Swinderen, 2015*). Although sleep may serve many diverse functions, and although many different molecular processes might be involved, in most animals sleep is understood to result largely from widespread changes in electrical activity throughout the brain. That is, the rapidly reversible switch that allows the aforementioned potential functions to take place probably requires important changes in electrical activity in the brain: initially to decrease behavioral responsiveness and thereby allow some of these processes to proceed without ongoing behavioral interference (*Mednick et al., 2011*).

In mammals, dis-inhibition of GABAergic sleep-promoting neurons in the vasolateral preoptic nucleus (VLPO) is thought to comprise part of a sleep switch (*Morairty et al., 2004*; *Saper et al.,*

*2010*). Interestingly, even the smallest brains appear to have a sleep switch: in the fruit fly *Drosophila melanogaster*, the dorsal fan-shaped body (dFB) has been suggested to play a role that is analogous to that of the mammalian VLPO (*Donlea et al., 2014, 2018*; *Pimentel et al., 2016*). Indeed, activation of these neurons in the fly's central brain promotes sleep and achieves key sleep functions (*Dissel et al., 2015*; *Donlea et al., 2011*). Recently, we have shown that changes in electrical activity in the fly brain are associated with spontaneous sleep transitions, and that activation of dFB neurons causes specific oscillatory signatures in local field potential (LFP) activity recorded from the central brain (*Yap et al., 2017*). It is not clear, however, how dFB neurons might communicate this activity to the rest of the fly brain to promote sleep rapidly. In addition, dFB-induced sleep has primarily been studied by observing increased periods of behavioral quiescence, rather than behavioral responsiveness to external stimuli. Decreased behavioral responsiveness would appear to be a prerequisite for sleep induction, but it remains unclear whether the same neurons could be involved in both acute loss of responsiveness and promotion of sleep functions (*Lebestky et al., 2009*), and if so, how the dFB might control both processes simultaneously. While it is understood that sleep is associated with decreased behavioral responsiveness, it is unclear whether the *Drosophila* dFB plays a larger role in regulating arousal more generally, even in awake animals. Recent research identified a neurochemical mechanism for promoting sleep in *Drosophila* (*Donlea et al., 2018*), but it is unknown whether other pathways are employed to link effectively internal sleep pressure signals in the dFB with the acute need to suppress behavioral responsiveness when flies need to sleep.

In addition to neurochemical signaling from sleep centers such as the VLPO, mammalian sleep processes might also involve electrical signaling that is mediated via gap junctions (*Coulon and Landisman, 2017*; *Franco-Pérez and Paz, 2009*). Gap-junction-mediated communication appears to be important for the rapid recruitment of much of the mammalian brain into synchronously firing networks (*Bennett and Zukin, 2004*; *Buzsaki, 2006*), but it remains unknown whether this is an important aspect of sleep physiology and function. In vertebrates, connexin genes code for a variety of gap junction subtypes (*Söhl and Willecke, 2004*). Invertebrates such as flies express a family of gap-junction genes that encode proteins called innexins, and *Drosophila* has eight innexin-encoding loci that have been implicated extensively in the development of the brain and other tissues (*Bauer et al., 2005*; *Stebbings et al., 2002*). In the adult fly, there is limited understanding of the role of gap-junction signaling in behavior, but studies have found a role for this signaling in visual processing (*Cuntz et al., 2007*; *Liu et al., 2016a*), in escape behavior (*Phelan et al., 1996*) and in learning and memory (*Wu et al., 2011*). In this study, we use optogenetics and electrophysiology to investigate the role of the dFB neurons in regulating behavioral responsiveness alongside sleep in the *Drosophila* model. We then examine how electrical and behavioral readouts of our sleep switch manipulations are affected when we remove *innexin6* gap junctions from the dFB.

## Results

### Correlating sleep duration and behavioral responsiveness

Flies were filmed in the *Drosophila* ARousal Tracking (DART) platform (*Figure 1A*) (*Faville et al., 2015*) to monitor sleep duration and behavioral responsiveness simultaneously (*Figure 1B*). Sleep duration is measured by well-established inactivity criteria based on >5 min inactivity (*Figure 1B*, upper panel; *Figure 1C*) (*Shaw et al., 2000*), whereas behavioral responsiveness can be measured by tracking how flies respond to a mechanical stimulus, during both sleep or while they are awake (*van Alphen et al., 2013*). Following a vibration stimulus, responding flies typically increase their locomotion (*Figure 1B*, lower panel). A stimulus that is delivered hourly provides an estimate of behavioral responsiveness throughout the circadian cycle (*Figure 1D*), and average responses (mean peak responsiveness, see 'Materials and methods') are typically stronger during the day than during the night (*Figure 1E*, left and middle panel). Responsiveness metrics therefore complement sleep duration measures (*Figure 1E*, right): animals are by definition less responsive when they are asleep (*Campbell and Tobler, 1984*).

Sleep duration and behavioral responsiveness should be negatively correlated. However, reduced behavioral responsiveness is also a feature of wakefulness, and animals that sleep more are not necessarily less responsive in general, even while awake. To better understand the relationship between these distinct arousal measures, we tracked average sleep duration and average responsiveness in

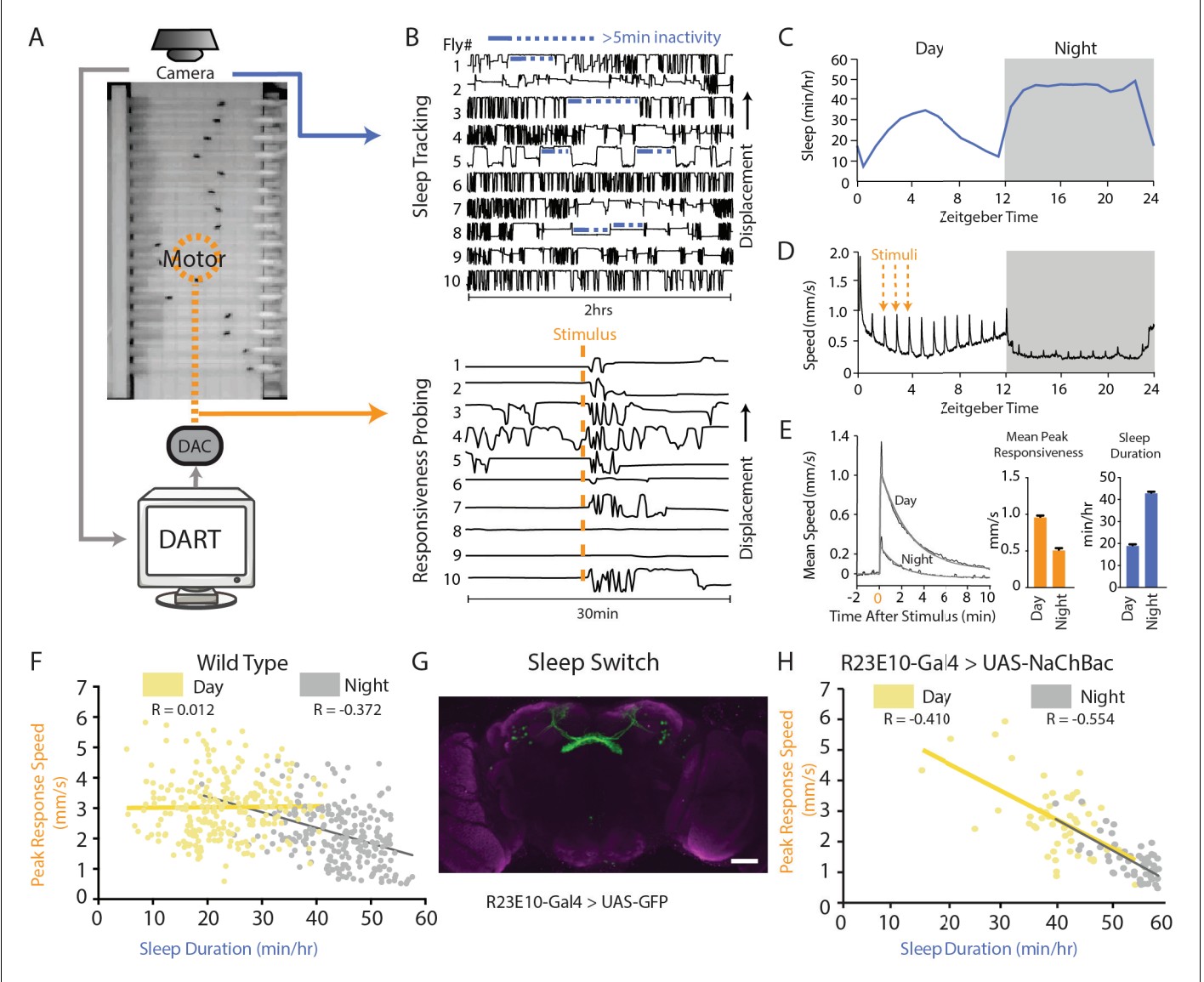

**Figure 1.** The sleep switch modulates behavioral responsiveness. (**A**) Flies in glass tubes were filmed from above and DART was used to track activity and to test behavioral responsiveness using a mechanical vibration. (**B**) Sleep duration was measured using 5 min inactivity criteria (top panel). Behavioral responsiveness was tested by quantifying the change in fly activity following a vibration stimulus. Following the stimulus (orange line), flies increase their locomotion speed as shown by their displacement in the tube (bottom panel). (**C**) Mean sleep duration (min/hr) is tracked over a circadian cycle. (**D**) Fly activity (speed, mm/sec) is plotted for a 24-hr day/night (white and grey, respectively) cycle during which a five-pulse 0.2 s 2.4 g vibration is delivered once per hour. Spikes in activity show timing of the stimuli, and the orange lines highlight three examples. (**E**) The mean response (speed, mm/s) for all stimuli during the day or night (left panel, black line). Shown in grey is a fitted curve for this average response (see 'Materials and methods'), the peak of which is a measure of the magnitude of response to the stimulus (middle panel). Responsiveness is greater during the day and lower during the night, whereas sleep duration is decreased during the day and increased at night (right panel). (**F**) Correlation between the peak response speed (mm/s) and sleep duration (min/hr) for wildtype (w2202) flies (n = 225) during the day (yellow R = 0.012, p=0.84) and the night (grey R = −0.372, p<0.0001). (**G**) R23E10-Gal4 neurons were chronically activated by expressing NaChBac, a bacterial sodium channel. Scale bar = 50 μm, the genotype in this image is R23E10-Gal4/+;UAS-2eGFP/+. (**H**) Correlation between responsiveness and sleep duration following activation of the dFB (R23E10-Gal4/UAS-NaChBac, n = 51) during daytimes (yellow R = −0.410, p<0.001) and nighttimes (grey R = −0.554, p<0.0001). See also *Figure 1— figure supplement 1*.

DOI: https://doi.org/10.7554/eLife.37105.002

The following figure supplement is available for figure 1:

**Figure supplement 1.** Correlation between responsiveness and sleep duration following activation of the dFB in various Gal4 drivers during the day (yellow) and night (grey).

DOI: https://doi.org/10.7554/eLife.37105.003

250 inbred wild-type flies (w[2202] [*Faville et al., 2015*]) over multiple days and nights. Locomotion immediately following the hourly vibrations (in awake or sleeping flies) contributed to the responsiveness metrics, and sleep duration was averaged from unstimulated epochs in the intervening hours. In previous work, we have shown that such hourly probing is not sleep depriving in wildtype flies (*van Alphen et al., 2013*). We found a surprisingly large amount of individual variability for these responsiveness and sleep metrics (*Figure 1F*). Sleep duration and behavioral responsiveness were only negatively correlated at night, but not during the day in the wildtype flies (*Figure 1F*). While this could reflect a qualitative difference between daytime sleep and nighttime sleep (*Faville et al., 2015*), or might indicate that sleep bouts are generally longer at night (*Shaw et al., 2000*), this also suggests that sleep duration and responsiveness might be separately controlled, as has been proposed by the authors of a previous study (*Lebestky et al., 2009*). Sleep-promoting circuits might also regulate responsiveness to external stimuli.

The dFB (*Figure 1G*) has been identified as a 'sleep switch' in the fly brain (*Donlea et al., 2011, 2014*; *Pimentel et al., 2016*), but its role in controlling behavioral responsiveness has not been well studied. dFB activation is associated with increased arousal thresholds (*Donlea et al., 2011, 2014*; *Pimentel et al., 2016*), but it is unclear whether this is a direct feature of increased dFB activity or a consequence of other processes that result from increased dFB activity. We first examined whether sleep duration and behavioral responsiveness might be simultaneously modulated by these neurons. Previous work has focussed on three Gal4 drivers that induce dFB-driven sleep in flies: 104y-Gal4, C5-Gal4, and R23E10-Gal4 (*Donlea et al., 2011, 2014*; *Pimentel et al., 2016*). We activated these sleep-promoting circuits by expressing a bacterial sodium channel, NaChBac (*Nitabach et al., 2006*). Chronic activation of the R23E10 neurons produced a strong negative correlation between sleep duration and behavioral responsiveness, even for daytime sleep (*Figure 1H*): flies that slept more tended to respond less to stimuli, day or night. Genetic controls showed only a night-time correlation (*Figure 1—figure supplement 1A*), as seen in the wildtype background strain (*Figure 1F*). In contrast to data from R23E10, results with C5 and 104y were less clear: C5 displayed a daytime correlation but not the nighttime correlation, and 104y did not display a daytime correlation (*Figure 1—figure supplement 1B,C*). This suggests that R23E10, which is a cleaner dFB driver (*Jenett et al., 2012*), may be more effective as a sleep switch. The R23E10 result also suggests that the dFB might regulate behavioral responsiveness as well as sleep duration, although it is difficult to separate these arousal measures in chronic manipulations.

## Optogenetic activation of the sleep switch decreases behavioral responsiveness

Falling asleep is an acute event that happens within seconds in most animals (*Campbell and Tobler, 1984*), and even flies can be significantly less responsive after only one minute of spontaneous quiescence (*van Alphen et al., 2013*; *Faville et al., 2015*). Optogenetic tools in *Drosophila* provide one way to induce sleep on demand experimentally. We used the R23E10-Gal4 driver to express CsChrimson, a red-light shifted channelrhodopsin (*Klapoetke et al., 2014*), in order to activate dFB neurons transiently in fly populations monitored by DART. As sleep is necessarily measured over prolonged time periods, we first examined whether prolonged optogenetic activation of the dFB was associated with decreased behavioral responsiveness. We therefore measured sleep duration and behavioral responsiveness in R23E10-Gal4 > UAS CsChrimson flies before, during, and after 12 hr of daytime dFB activation (*Figure 2A,C*). All-trans retinal (ATR) was fed to flies to enable channelrhodopsin function, and genetically identical but non-ATR-fed flies were used as controls (*Figure 2A*, black). Flies that were fed ATR significantly increased their sleep duration in response to dFB activation with prolonged red light exposure (*Figure 2A,B*, blue). Behavioral responsiveness was robust in these flies at baseline (*Figure 2C*, left panel), but optogenetic activation of the dFB significantly decreased responsiveness to the vibration stimulus (*Figure 2C*, right panel; *Figure 2D*, left panel). Responsiveness was not, however, abolished by activation of the dFB, and even appeared to recover somewhat after several hours of red-light exposure (*Figure 2C*, right panel).

So far, our behavioral responsiveness metric does not differentiate between waking or sleep in flies. When responsiveness is analyzed only for sleeping flies, expressed relative to the proportion of immobile flies that respond (at any level) to the stimulus, this provides a measure of sleep intensity (*Figure 2—figure supplement 1A*). Wildtype flies sleep more deeply at night than during the day (*Figure 2—figure supplement 1B,C*), and sleep intensity is not necessarily correlated to sleep

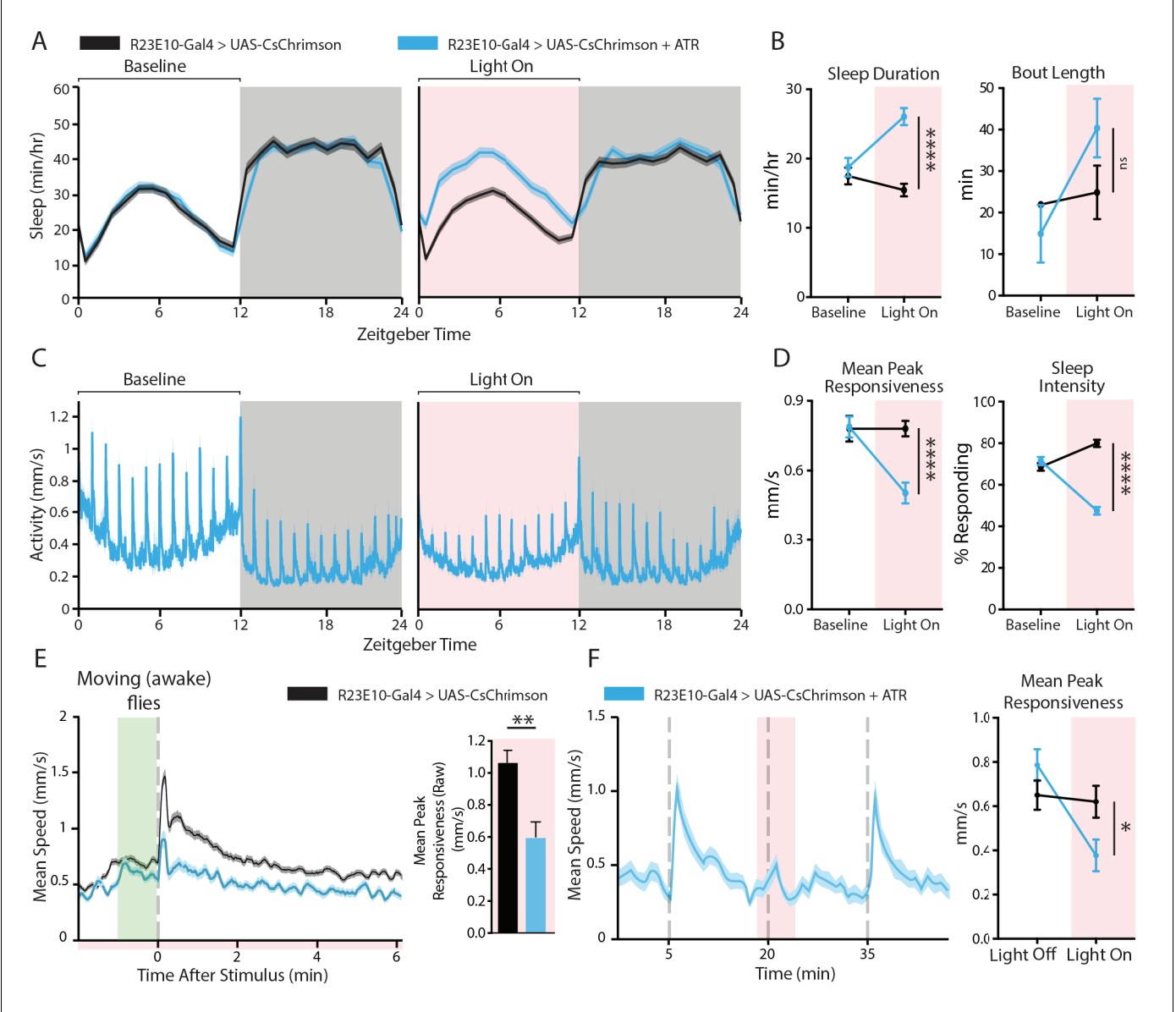

**Figure 2.** Behavioral effects of acutely activating R23E10 neurons. (A–D) Effects on sleep and responsiveness following the activation of R23E10 neurons (blue, UAS-CsChrimson/+;R23E10-Gal4/+ with ATR, n = 102) compared to those in control flies (black, no ATR feeding, n = 118). Error bars and shading indicate standard errors of the mean (SEMs) and asterisks indicate significance (****p<0.0001, ns = not significant, t-tests). (A) Mean sleep duration (min/hr) during the period 24 hr before red light activation (left), then during the next 24 hr when red light is delivered for 12 hr during the day (pink shading, right). (B) Comparison of the 12-hr day period without red light (baseline) to the period of red-light activation in terms of sleep duration and bout length. (C) Mean activity (mm/s) for R23E10 activation for the time periods in (A). Spikes in activity show timing of hourly vibration stimuli. (D) Comparison between the 12-hr day period without red light (baseline) and the period of red-light activation for peak responsiveness and sleep intensity. (E) Left, average stimulus response for UAS-CsChrimson/+;R23E10-Gal4/+ with ATR (blue, n = 50) compared to control flies (black, no ATR feeding, n = 48) during red-light activation (*Figure 2A–D* red shading) in flies that moved in the minute prior to the stimulus (i.e. awake flies). Right, summary histogram (average ± SEM). **p<0.01, t-test. (F) Example activity trace of flies responding to stimuli 15 min apart (gray dashed lines): 1 min CsChrimson activation (red shading) prior to the stimulus is alternated with trials without red light (left panel). One minute of dFB activation is sufficient to decrease responsiveness (right panel, UAS-CsChrimson/+;R23E10-Gal4/+ with ATR n = 83, control n = 80, *p<0.05, t-test). See also *Figure 2—figure supplements 1*, *2* and *3*.

DOI: https://doi.org/10.7554/eLife.37105.004

The following figure supplements are available for figure 2:

**Figure supplement 1.** Measuring sleep intensity.

DOI: https://doi.org/10.7554/eLife.37105.005

*Figure 2 continued on next page*

*Figure 2 continued*

**Figure supplement 2.** Acute effects in awake flies.
DOI: https://doi.org/10.7554/eLife.37105.006
**Figure supplement 3.** 1 Hz optogenetic activation of the dFB.
DOI: https://doi.org/10.7554/eLife.37105.007

duration (*van Alphen et al., 2013*; *Faville et al., 2015*). Yet, as might be expected, optogenetic activation of the 'sleep switch' increases sleep intensity (*Figure 2D*, right panel; *Figure 2—figure supplement 1D*) as well as sleep duration. However, dFB-activated flies are not always asleep, they are often awake. We therefore next investigated whether responsiveness was affected in awake (i.e. walking) flies only (see 'Materials and methods' and *Figure 2—figure supplement 1A* for explanations on how wakeful responsiveness was determined). We were surprised to find that dFB activation decreases behavioral responsiveness in awake flies as well as in sleeping flies (*Figure 2E*, right panel). Thus, decreased behavioral responsiveness appears to be a feature of prolonged dFB activation, even when sleep reverts to wakefulness. Decreased responsiveness was not due to sluggish locomotion in awake animals: the walking speed of awake dFB-activated flies was not different from that of controls immediately preceding the vibration stimulus (*Figure 2E*, left panel, green shading). dFB activation therefore reduces behavioral responsiveness in awake as well as in sleeping flies.

To confirm that this loss of behavioral responsiveness in awake flies is an acute effect of dFB activation and not just an indirect effect of prolonged sleep induction, we repeated these experiments with shorter time periods (5 min) of dFB activation. We probed for responsiveness after only 1 min of red-light exposure. In doing so, we found that behavioral responsiveness was significantly reduced after 1 min of red light, and that responsiveness was rapidly restored after the red light was turned off (*Figure 2F*), that is we observed no evidence of sleep inertia following 5 min of dFB activation. To further confirm that this effect was distinct from sleep, we also analyzed only flies that were walking in the minute preceding the vibration stimulus, and observed the same significant effect (*Figure 2—figure supplement 2*). Together with our prolonged activation experiments, this suggests a dissociation between dFB's role in sleep promotion and behavioral responsiveness. While more prolonged dFB activation certainly increased sleep duration and sleep intensity (*Figure 2B,D*; *Figure 2—figure supplement 1D*), behavioral responsiveness remains suppressed when dFB-activated flies are awake and can be acutely suppressed in walking flies. As a constant light activation might be an unnatural stimulation regime for the dFB neurons, we repeated our experiments with a sparser activation stimulus, pulsed 5 ms red light (1 Hz), and found that behavioral responsiveness was similarly suppressed (*Figure 2—figure supplement 3*). This shows that different stimulation regimes produce a similar effect and confirms that activation of the sleep switch suppresses behavioral responsiveness.

## Optogenetic activation of the sleep switch causes a rapid and reversible increase in membrane potential of dFB neurons

To better understand the mechanisms underlying these acute effects, we investigated the electrophysiological properties of the dFB neurons that contributed to the observed change in behavior. We used whole-cell patch-clamp electrophysiology in current clamp mode to record the membrane potential of the dFB cell bodies in behaving flies (*Figure 3A*) (see 'Materials and methods'). dFB neurons were genetically labeled with green fluorescent protein (GFP) to guide recordings from the cell soma visually (*Figure 3B*, top). As shown previously (*Donlea et al., 2014*; *Pimentel et al., 2016*), the spiking activity of dFB soma responded to positive current steps by increasing their firing rate (*Figure 3B*, bottom left). However, not all neurons shared the same spontaneous firing properties, with ~20% of the recorded cells identified as non-spiking (even following current injection), whereas the remaining cells that did spike were observed to exhibit various combinations of single spikes and burst spiking patterns (*Figure 3B*, bottom right).

Despite the heterogeneity in the firing pattern of the dFB neurons, CsChrimson-expressing cells all responded to a red-light stimulus in the same manner, by rapidly increasing their membrane potential upon activation, followed by a gradual return to its resting membrane state after the light was turned off (*Figure 3C,D*). Induced spike bursts occurred 18.75 ± 3.07 ms apart in a subset of cells. Exposure to 1 Hz light pulses (5 ms each) instead of continuous exposure produced corresponding action

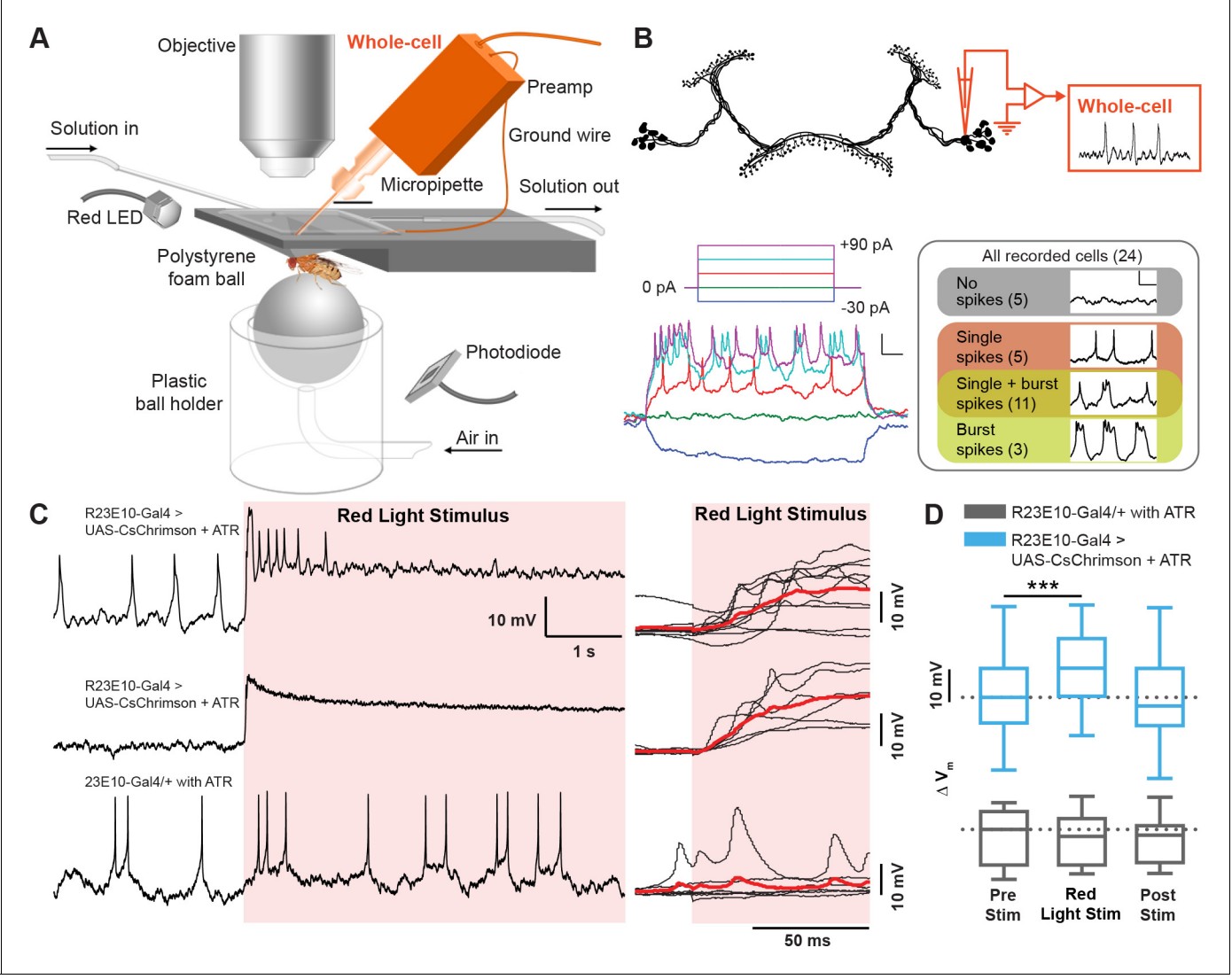

**Figure 3.** Electrophysiological effects of acutely activating R23E10 neurons. (A) Setup for recording in vivo adult *Drosophila* electrophysiology with whole-cell patch clamp (orange). (B) dFB neuron schematic showing whole-cell recordings targeted to R23E10-Gal4 cell bodies. Injecting current in a stepwise manner causes firing in these neurons (bottom left panel), which are heterogeneous in their endogenous firing patterns (bottom right panel). Scale bars indicate 10 mV and 100 ms. (C) Example traces (left) of a CsChrimson-expressing (UAS-CsChrimson/+;R23E10-Gal4/+ with ATR) spiking cell (top) and non-spiking cell (middle), and of a non-CsChrimson-expressing (R23E10-Gal4/+ with ATR) cell (bottom) when exposed to constant red light (red shading). Superimposed traces of the corresponding types from multiple cell recordings (right panel, top to bottom: n = 10, n = 6, n = 6). Solid red lines indicate mean values. (D) Boxplots show median membrane potentials for CsChrimson-expressing cells (blue) and non-CsChrimson expressing cells (gray, n = 6) before, during and after constant light stimulation (***p<0.001, Friedman test with Dunn's multiple comparisons to pre-stimulus condition). See also *Figure 3—figure supplements 1* and *2*.

DOI: https://doi.org/10.7554/eLife.37105.008

The following figure supplements are available for figure 3:

**Figure supplement 1.** 1Hz stimulation.

DOI: https://doi.org/10.7554/eLife.37105.009

**Figure supplement 2.** Whole-cell patch physiology in wildtype and INX6 knockdown flies.

DOI: https://doi.org/10.7554/eLife.37105.010

potentials in some spiking cells (*Figure 3—figure supplement 1A*). On average, 1 Hz pulsed activation also increased the membrane potential in all CsChrimson-expressing dFB cells (*Figure 3—figure supplement 1B*). The robustness of this effect, despite the heterogeneity in the spontaneous firing pattern across cells and the different stimulus regimes, suggests that the acute dFB activation effects

on behavioral responsiveness could be associated with increased membrane potential rather than with any specific increased spiking activity, although it remains possible that the subset of spiking cells is key here. Notably, the post-stimulation membrane potential (recorded only 1 min after the red light was turned off) was not significantly different from baseline (*Figure 3D*; *Figure 3—figure supplement 1B*), indicating that the acute effect on the membrane potential was not sustained for long after the stimulation. Red light had no effect on spiking or the membrane potential in R23E10-Gal4/+ animals that had been fed ATR but that lacked channelrhodopsin (*Figure 3D*, *Figure 3—figure supplement 1B*, gray boxplots), or in control R23E10-Gal4/UAS-CsChrimson animals that had not been fed ATR (*Figure 3—figure supplement 2*). In a subset of recordings, 1 Hz light pulses reliably evoked secondary and even tertiary spikes between 100 and 500 ms after the first evoked spike (*Figure 3—figure supplement 2*), suggesting a reverberation in the circuit.

## Gap-junction localization in the dFB

An increased input resistance and membrane time constant in dFB cells has been shown to be associated with increased sleep pressure (*Donlea et al., 2014*), so our optogenetic results are consistent with the idea that these cells form part of a sleep homeostat that is regulated by changing membrane potential levels (*Pimentel et al., 2016*). Recent work shows that output from the dFB cells regulates sleep duration by inhibiting a number of other systems in the central brain (*Donlea et al., 2018*; *Liu et al., 2016b*). These communication channels are probably chemical in nature, but it is not clear whether behavioral responsiveness is also controlled through these downstream synaptic circuits. We therefore next acutely manipulated synaptic release from the dFB. Interestingly, transiently increasing synaptic activity in dFB neurons (measured using temperature-sensitive *syntaxin*$^{3-69}$ [*Lagow et al., 2007*; *Kottler et al., 2013*]) had no significant effect on behavioral responsiveness, or on sleep (*Figure 4—figure supplement 1A*). Transiently decreasing synaptic activity (by using the temperature-sensitive *shibire* [*Kitamoto, 2001*]) in the dFB decreased responsiveness, but equally so in the genetic controls (*Figure 4—figure supplement 1B*). A lack of clear effects using these opposing synaptic manipulations suggests that (fast) chemical neurotransmission might not be a relevant mechanism employed by these neurons to control behavior acutely, or at least that these acute synaptic manipulations do not affect our behavioral readouts. This prompted us to explore whether electrical synapses might be involved instead. A previous study has found expression of the gap junction gene, inx6, in the fan-shaped body (*Wu et al., 2011*), suggesting that these neurons might communicate electrically with other neurons potentially to regulate behavioral responsiveness.

We used an antibody for the INX6 protein to first visualize its location in the fly brain. Expression was seen most strongly in a dorsal layer of the fan-shaped body and in large cells in the pars intercerebralis (PI) (*Figure 4A*, top and middle panels, *Figure 4—figure supplement 2A*). Co-labeling with GFP driven by R23E10-Gal4 revealed a significant overlap in the relevant dFB neurons (*Figure 4B*, bottom panel; a co-localization of 47.5% was found, see 'Materials and methods'). However, we did not find any evidence of co-localization in the dFB cell soma (*Figure 4—figure supplement 2B*). Dye-fill experiments confirmed that the dFB neurons are coupled to other cells via gap junctions (*Figure 4—figure supplement 2C*; *Video 1*), although the extent of this electrically coupled network remains unclear as only the large PI cells above the dFB revealed any reliable dye coupling in our experiments.

## Gap junctions in the dFB regulate both sleep duration and behavioral responsiveness

To test the role of INX6 in sleep, we used a working RNAi construct (*Figure 5—figure supplement 1A,B*) to knock down INX6 in dFB neurons. Flies expressing the INX6 RNAi construct in the dFB showed a decrease in both day and night sleep (*Figure 5A*). We next looked at responsiveness to mechanical stimuli in these flies. Flies slept less deeply and were more responsive to stimuli when INX6 was downregulated in the dFB (*Figure 5B*). Closer examination of sleep intensity as a function of time asleep (*van Alphen et al., 2013*) showed that INX6-knockdown flies could still achieve deeper sleep stages (e.g. after 16–20 min immobility, *Figure 5C*), although these flies slept more lightly in general. This suggests that INX6 plays a role in suppressing behavioral responsiveness during lighter sleep stages, whereas during deeper sleep, the role of INX6 might be less important. Circadian influences also clearly play a major role: responsiveness is always lower at night than during

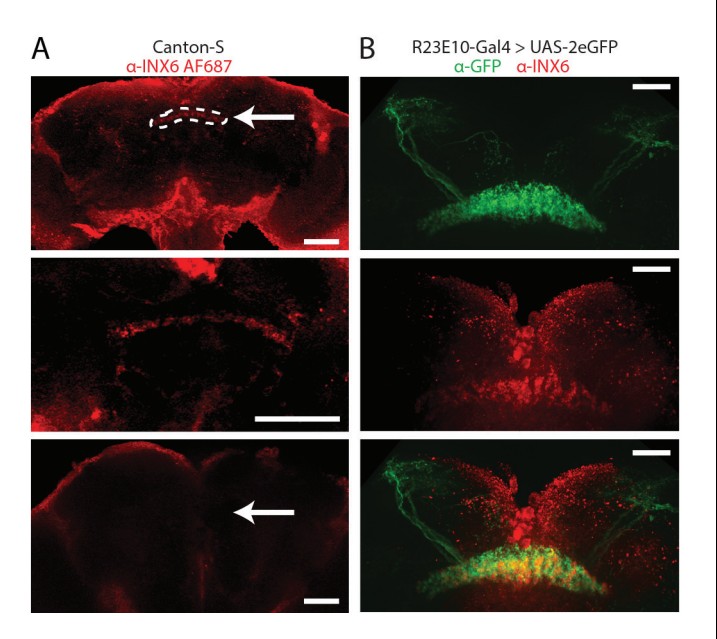

**Figure 4.** INX6 dFB localization. (**A**) INX6 antibody staining (red) in wildtype Canton-S flies at 20x (top) and 60x (middle). White arrow indicates the location of the dFB and white dashes outline it. Staining using the secondary antibody alone (bottom, 20x) shows no detectable reactivity in the dFB. Scale bars = 50 μm. (**B**) INX6 antibody staining (red, middle) with GFP antibody staining in flies expressing GFP in R23E10 neurons (UAS-GFP/+;R23E10-Gal4/+, green, top). Overlap between these regions (bottom) indicates the presence of INX6 in these neurons (47.5% co-localization, see 'Materials and methods'). Scale bars = 25 μm. See also *Figure 4—figure supplements 1* and *2*.

DOI: https://doi.org/10.7554/eLife.37105.011

The following figure supplements are available for figure 4:

**Figure supplement 1.** Acute synaptic manipulations.

DOI: https://doi.org/10.7554/eLife.37105.012

**Figure supplement 2.** INX6 in dFB neurons.

DOI: https://doi.org/10.7554/eLife.37105.013

the day. In addition, INX6 expression in different subsets of dFB neurons could also contribute to the behavior, as we observed only partial INX6 overlap with R23E10 (*Figure 4B*, bottom). To confirm that these behavioral effects were not a consequence of altered developmental pathways, we downregulated INX6 in adult flies by using a temperature-sensitive suppressor of Gal4, Gal80$^{TS}$ (*McGuire et al., 2004*). Knocking down INX6 in the adult dFB significantly increased daytime behavioral responsiveness and decreased daytime sleep intensity, but had no significant effect on sleep duration (*Figure 5—figure supplement 2*). These results suggest that INX6 is important for communicating dFB activity levels to regulate behavioral responsiveness. We therefore next questioned whether INX6 knockdown altered electrical readouts associated with acute dFB activation.

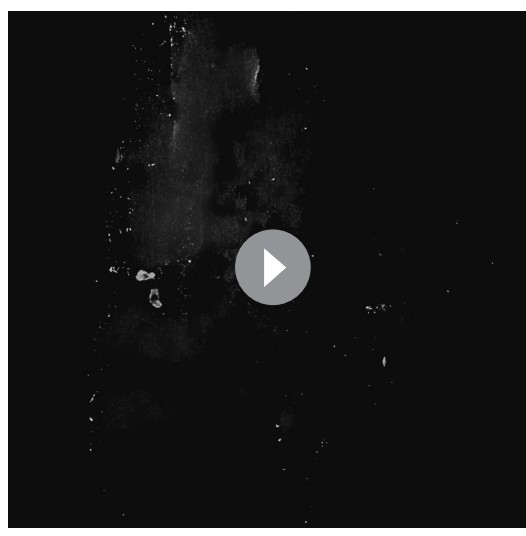

**Video 1.** 3D reconstruction of dye coupling experiment.

DOI: https://doi.org/10.7554/eLife.37105.014

## Gap junctions promote LFP activity in the dFB

We recorded neural activity from INX6 knockdown flies whose dFB could be optogenetically activated. A dual-electrode setup was used to record intracellularly from the dFB cell body while a second electrode simultaneously positioned within in the dFB structure recorded extracellular local field potentials (LFPs) (*Figure 6A,B*). Whole-cell recordings revealed that reduced INX6 expression in the dFB (*Figure 5—figure supplement 1B*) did not impair CsChrimson-mediated cell activation: a significant increase in membrane potential was still observed with constant red light as well as with 1 Hz pulses (*Figure 6C,D*). This shows that the increase of the membrane potential in activated dFB neurons does not depend upon INX6 expression in these cells. Nevertheless, we did observe some intracellular differences: whereas 40% of wildtype dFB cells exhibited reliable secondary spikes 100–500 ms after a light-evoked action potential (*Figure 3—figure supplement 2*), none of the INX6 knocked-down cells displayed secondary spikes within that timeframe (*Figure 6—figure supplement 1*). Also, we noted considerably less variability in the membrane potential for dFB cells lacking INX6 (*Figure 6—figure supplement 1* and *Figure 6C,D* right panels, compared with *Figure 3D* and *Figure 3—figure supplement 1B*).

We next asked whether extracellular LFP activity in the dFB might be affected by INX6 knockdown in the R23E10 neurons. We have shown previously that optogenetic dFB activation using the same R23E10-Gal4 driver produces increased LFP activity in the fly brain, especially in low (1–15 Hz) frequencies, and that oscillations within this range are also seen during transitions to spontaneous sleep in flies (*Yap et al., 2017*). We find this LFP signature of dFB activation again here (*Figure 6E*, top panel; *Figure 6F*, blue). In flies with INX6 knocked down in the dFB, however, there was no increased LFP activity (*Figure 6E*, bottom panel; *Figure 6F*, purple), as was also the case for control flies lacking the channelrhodopsin (*Figure 6F*, grey). As this constant light stimulus effect was quite subtle, we decided to drive the system with a pulsed activation regime to further test whether field potentials in the dFB were compromised following INX6 knockdown. 1 Hz stimulation of R23E10-Gal4 neurons also produced an LFP signature in the dFB, evident as an event-related potential (ERP) (*Figure 6G*, blue). Consistent with our results using a constant red-light activation regime, INX6 knockdown in R23E10 neurons significantly blunted the amplitude of the 1 Hz ERP (*Figure 6G*, purple). Together, these results suggest that one important consequence of INX6 knockdown in the sleep-promoting R23E10 cells is decreased synchronous activity in the dFB (detected as LFPs), whereas the activation effects on the membrane potential measured at the soma remain robust. To further test whether gap junctions are involved, we applied the gap-junction blocker carbenoxolone, or CBX (*Cao and Nitabach, 2008*). Perfusion of CBX onto the brain blunted the ERP recorded from the dFB, as did locally injected CBX (*Figure 6H*). This adds further evidence that gap junctions are required to produce synchronized activity in the dFB.

## Gap junctions in the dFB are required for controlling behavioral responsiveness

Given that INX6 knockdown appears to disrupt dFB function at an electrophysiological level, we hypothesized that removing INX6 from these neurons would impair acute effects on behavior, such as those we obtained by optogenetic approaches earlier (*Figure 2*). To test this, we compared three groups of flies in DART (*Figure 7A*): (1) Positive control flies where dFB neurons could be activated and had wildtype INX6 expression (ATR-fed, blue), (2) negative control flies with INX6 knocked down but without dFB activation (non-ATR-fed, black), and (3) genetically identical flies where dFB neurons could be optogenetically activated but with INX6 knocked down (ATR-fed, purple).

Optogenetically activating the dFB in cells where INX6 was knocked down still caused an increase in sleep duration and bout length compared to those in negative control flies (*Figure 7A,B*, purple compared to black). This effect was, however, not as strong as the activation of the dFB circuit with functional INX6 in these cells (*Figure 7A,B*, blue). This shows that INX6 knockdown blunts the sleep-promoting effect of dFB activation. Correspondingly, behavioral responsiveness was significantly decreased when the dFB was activated whereas INX6 expression was intact (*Figure 7C,D* blue). We did not see differences in peak responsiveness at baseline in the knockdown strains (*Figure 7D*, left), although the baseline sleep intensity data were consistent with our previous findings that inx6 knockdown in the dFB increases responsiveness (*Figure 7D*, right). It is possible that CsChrimson activity is leaky in ambient light conditions. However, clear differences were evident upon red-light activation: in

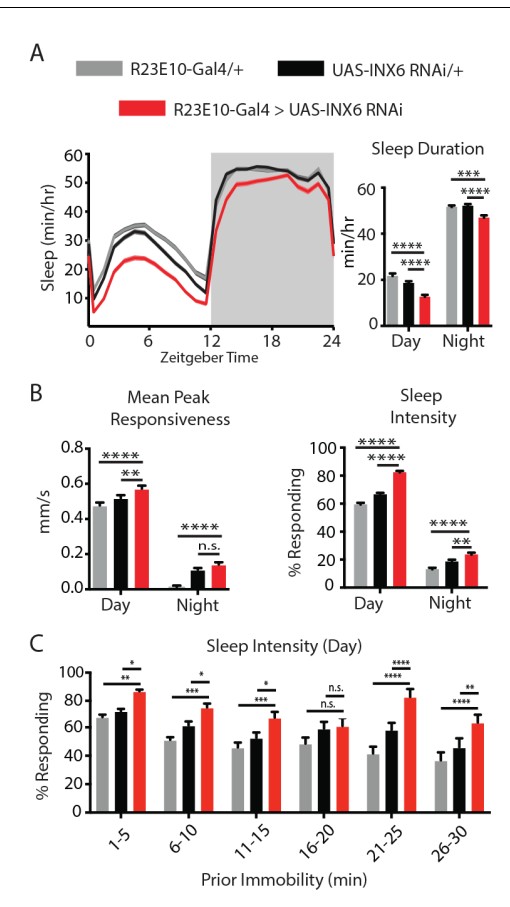

**Figure 5.** INX6 effects on behavior. (**A**) Mean sleep duration (min/hr) and (**B**) mean peak responsiveness (mm/s) and sleep intensity (% responding) for R23E10-Gal4/+;UAS-INX6-RNAi/+ (red, n = 85) compared to R23E10-Gal4/+ (gray, n = 83) and UAS-INX6-RNAi/+ (black, n = 84) controls. (**C**) Sleep intensity for inactivity bins of 5 min for the first 30 min of inactivity. See also *Figure 2—figure supplement 1*. Shading and error bars indicate SEM, asterisks indicate significance (*p<0.05, **p<0.01, ***p<0.001, ****p<0.0001, two-way ANOVA, adjusted for multiple comparison (Dunnett). See also *Figure 5—figure supplements 1* and *2*.
DOI: https://doi.org/10.7554/eLife.37105.015

The following figure supplements are available for figure 5:

**Figure supplement 1.** INX6 RNAi effectiveness.
DOI: https://doi.org/10.7554/eLife.37105.016

**Figure supplement 2.** Behavioral effects of acute downregulation of INX6 in adult flies.
DOI: https://doi.org/10.7554/eLife.37105.017

contrast to the partial effects on sleep duration, INX6 knockdown completely blocked the effect of dFB activation on behavioral responsiveness (*Figure 7C*, purple) measured in terms of either mean peak responsiveness and sleep intensity (*Figure 7D*, purple), making the knockdown indistinguishable from negative controls (*Figure 7D*, black; not shown in *Figure 7C* for clarity). The knockdown was also indistinguishable from negative controls when we only examined data from awake flies, confirming that this effect is not restricted to sleep (p=0.94, *t*-test comparing mean responsiveness, n = 51 flies). These activation experiments therefore uncover a dissociation between behavioral responsiveness and sleep duration: INX6 expression is required for the sleep switch to downregulate behavioral responsiveness acutely, but INX6 is not required for these neurons to be sleep promoting. Presumably, dFB neurons are still able to affect sleep duration via other mechanisms (*Donlea et al., 2018*; *Liu et al., 2016b*).

## Discussion

Before sleep can begin to achieve any of its multiple putative functions, it would seem that behavioral responsiveness to the outside world first needs to be reduced. This is a fundamental yet poorly understood aspect of sleep in all animals (*Campbell and Tobler, 1984*; *Cirelli, 2009*). Mechanisms that regulate behavioral responsiveness are therefore likely to be involved during transitions from wakefulness to sleep, so it seems parsimonious that overlapping neuronal systems might govern both behavioral states. In this way, sleep onset has some similarity with selective attention in its capacity to gate or suppress perception rapidly (*Kirszenblat and van Swinderen, 2015*). We show here that gap junctions that are expressed in the sleep-promoting neurons of the dFB in the central fly brain are probably mediators for gating behavioral responsiveness in both awake and sleeping flies, as a consequence of acutely increased membrane potential in these neurons. Recent work has shown that sleep pressure increases the input resistance and membrane time constants in these same neurons (*Donlea et al., 2014*; *Pimentel et al., 2016*), so this seems to be a probable mechanism whereby increased sleep pressure impairs behavioral responsiveness via electrical channels, perhaps even before promoting and maintaining sleep — which may then occur via chemical (e.g. peptidergic) signaling from these neurons (*Donlea et al., 2018*). It is therefore possible that the R23E10 neurons, which have been recently characterized as comprising part of a 'sleep homeostat' in the fly brain (*Donlea et al., 2018*; *Pimentel et al., 2016*), have two distinct arousal-related functions: to regulate behavioral responsiveness more generally —

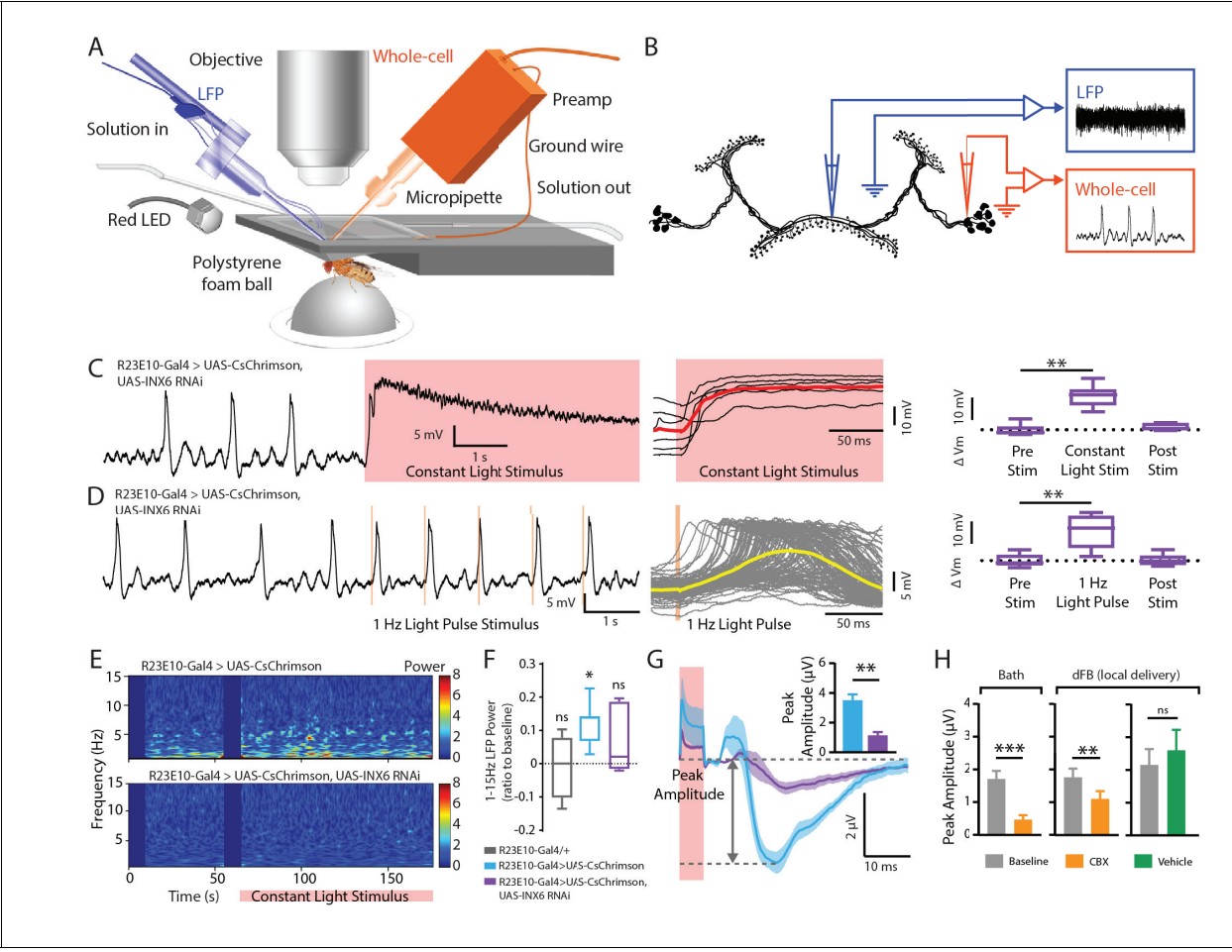

**Figure 6.** Electrophysiological effects of INX6 knockdown in activated R23E10 neurons. (**A**) In vivo adult *Drosophila* electrophysiology recording setup with a whole-cell patch clamp (orange) targeting R23E10 cell bodies and a local field potential (blue) electrode targeting the dFB. (**B**) Local field-potential (LFP) recordings were obtained within the presynaptic arborizations of the R23E10 neurons in the fan-shaped body, while whole-cell recordings were obtained from the cell bodies of R23E10 neurons. (**C**) Example trace (left) of a CsChrimson-expressing (UAS-CsChrimson/+; UAS-INX6 RNAi/+; R23E10-Gal4/+ with ATR) cell with INX6 knockdown when exposed to constant red light (red shading). Superimposed traces from multiple cell recordings (middle). The solid red line indicates mean values. Boxplots show median membrane potential for CsChrimson-expressing cells with INX6 knockdown before, during and after constant light stimulation (n = 6, **p<0.01, Friedman test with Dunn's multiple comparisons to the pre-stimulus condition). (**D**) Example trace (left) of a CsChrimson-expressing cell with INX6 knockdown when exposed to 1 Hz light pulse with 5 ms exposure per pulse (orange shading). Superimposed traces from all 120 trials (middle) for the representative cell shown on the left. Solid yellow line indicates mean value. Boxplots show median membrane potential for CsChrimson-expressing cells with INX6 knockdown before, during, and after 1 Hz light stimulation (n = 6, **p<0.01, Friedman test with Dunn's multiple comparisons to pre-stimulus condition). (**E**) Local field potential recordings from the dorsal fan-shaped body represented as power within the 1–15 Hz frequency range, in response to a constant light stimulus (pink bar) with one representative fly shown from each strain. (**F**) Boxplots show median 1–15 Hz local field potential power (±SEM) for the duration of the constant light stimulus relative to pre-stimulation power (normalized to zero) in UAS-CsChrimson/+;R23E10-Gal4/+ with ATR (blue, n = 7), UAS-CsChrimson/+; UAS-INX6 RNAi/+; R23E10-Gal4/+ with ATR (purple, n = 6), and R23E10-Gal4/+ with ATR (gray, n = 4). Only UAS-CsChrimson/+;R23E10-Gal4/+ showed a significant increase in 1–15 Hz activity when exposed to constant red light (*p<0.05, ns = not significant, by Wilcoxon signed rank test). (**G**) Peak amplitude of the average LFP response (±SEM) to 1 Hz light pulse stimulus is significantly reduced for UAS-CsChrimson/+; UAS-INX6 RNAi/+; R23E10-Gal4/+ (purple, n = 6), compared to UAS-CsChrimson/+;R23E10-Gal4/+ (blue, n = 7) (**p<0.01, by Mann-Whitney test). (**H**) Peak LFP amplitude (±SEM) in response to 1 Hz stimulus in the presence of bath-applied (n = 6) or locally injected (n = 4) carbenoxolone (CBX, orange) compared to baseline (gray) or vehicle (green, n = 5). ***p<0.001,**p<0.01, by paired t-test. See also *Figure 6—figure supplement 1*.

DOI: https://doi.org/10.7554/eLife.37105.018

The following figure supplement is available for figure 6:

**Figure supplement 1.** Combined traces (yellow lines indicate mean values) for all 120 pulses for each of the recorded CsChrimson-expressing R23E10 neurons with INX6 knockdown.

DOI: https://doi.org/10.7554/eLife.37105.019

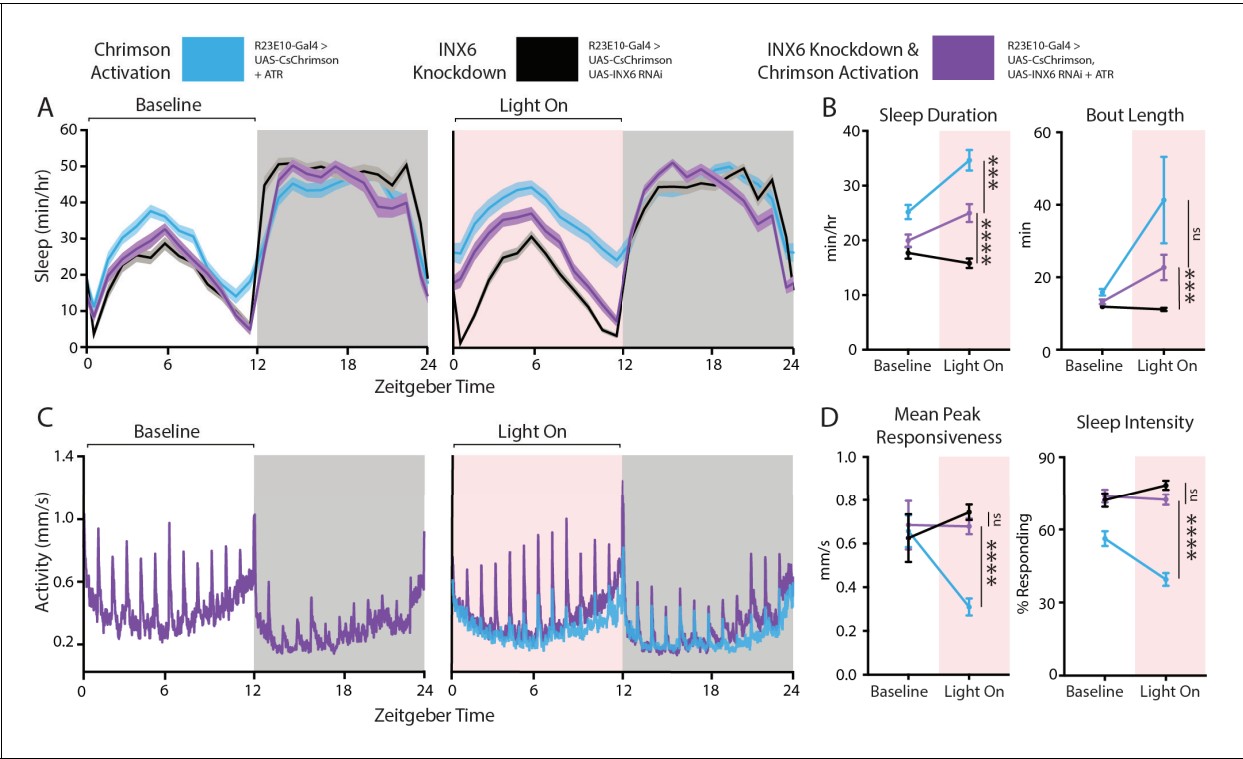

**Figure 7.** Behavioral effects of INX6 knockdown in activated R23E10 neurons. (**A–D**) Three different behavioral conditions comparing flies with R23E10 neurons that can be activated (UAS-CsChrimson/+; R23E10-Gal4/+with ATR, blue, n = 50) with flies where R23E10 neurons cannot be activated but have INX6 knocked down (UAS CsChrimson/+; UAS-INX6 RNAi/R23E10-Gal4 no ATR, black, n = 51) and flies with R23E10 neurons that can be activated and have INX6 knocked down (UAS-CsChrimson/+; UAS-INX6 RNAi/R23E10-Gal4 with ATR, purple, n = 51) for effects on sleep and responsiveness following red light activation. (**A**) Mean sleep duration (min/hr, shading indicates SEM) during the 24 hr before red light activation, then during the next 24 hr during which red light is delivered for 12 hr during the day (pink shading). (**B**) Comparison between the 12-hr day period without red light (baseline) and the period of red-light activation in terms of sleep duration and bout length. (**C**) Mean activity (mm/s) for UAS-CSChrimson/+; UAS-INX6 RNAi/R23E10-Gal4 with ATR for the same time periods as (**A**). UAS-CsChrimson/+; R23E10-Gal4/+ with ATR is overlaid (blue) during the 'Light On' period to show differences in responses to hourly stimuli. Other traces (black) not shown for clarity; see (**D**) for summarized control data. (**D**) Comparison between the 12-hr day period without red light (baseline) and the period of red-light activation in terms of peak responsiveness and sleep intensity. Error bars indicate SEM and asterisks indicate significance (***p<0.001, ****p<0.0001, t-tests).
DOI: https://doi.org/10.7554/eLife.37105.020

and thus to promote sleep acutely — and then to maintain sleep for the duration required by homeostatic demands. As increased arousal thresholds are a key criterion for sleep, it is of course difficult to disentangle these processes from each other. However, our data suggest that dFB-mediated suppression of behavioral responsiveness could be a prequel to sleep. First, on the basis of the current definitions of sleep (*Campbell and Tobler, 1984*; *Cirelli, 2009*), it does not make sense for sleep to precede changes in responsiveness, so either these processes are simultaneous or sleep functions succeed an initial increase in arousal thresholds that may already be evident during wakefulness.

Our experiments show that behavioral responsiveness can be suppressed by acutely activating the dFB even in awake flies, and that this effect has little inertia, at the levels of both behavior and electrophysiology. Sleep intensity was increased however upon prolonged dFB activation, suggesting a distinct cumulative effect. Indeed, at the deepest stage of sleep (when flies are least responsive to mechanical stimuli), removal of INX6 from the dFB had no effect on behavioral responsiveness (*Figure 5C*), suggesting that other factors are involved at that stage for maintaining high arousal thresholds. When flies are closer to awakening again, or when they are engaged in lighter sleep stages, it is possible that INX6-mediated mechanisms come back into play to maintain sleep. In recent work, we have shown that transitions in and out of sleep are associated with dFB activity (*Yap et al., 2017*), which is consistent with the view that a common system might be governing behavioral responsiveness during wakefulness as well as sleep.

It is likely that other dFB neurons outside of the R23E10 circuit are also involved, as suggested by only partial overlap with INX6 labeling (*Figure 4B*). However, without INX6, activated R23E10 neurons were unable to reduce behavioral responsiveness, suggesting a cell-autonomous effect. We have shown here and elsewhere (*Yap et al., 2017*) that dFB activation is associated with increased LFP activity in the central brain, which could reflect increased synchronous neural firing. Without expression of INX6 in the R23E10 neurons, these induced LFP effects also disappear, suggesting that gap junctions are important for promoting synchronous activity in this circuit. This suggests a rapid mechanism through which the dFB might suppress behavioral responsiveness upon sleep onset, by for example producing synchronous low frequency activity that could interfere with ongoing sensory processing and integration in the central brain. Although acute activation of the R23E10 neurons alone is unlikely to occur in a natural context, electrical communication to other cells via gap junctions could support subtler levels of behavioral control while the flies are awake, which might consolidate into overall increased arousal thresholds as membrane potentials across the dFB increase with increasing sleep pressure. We have not completely excluded the possibility that a fast-acting neuropeptide or neurotransmitter secreted from R23E10 neurons regulates behavioral responsiveness, but our data favor the view that a parallel channel involving electrical synapses exists. Future studies should uncover the extent of this gap-junction-coupled network, although our antibody-labeling (*Figure 4*) and dye-coupling experiments (*Figure 4—figure supplement 1*) suggest that cells in the pars intercerebralis (PI) are probably involved. It also remains possible that dFB neurons are electrically coupled to each other, thereby generating LFP oscillations that are employed in the regulation of behavior. However, in our dye-coupling experiments (see 'Materials and methods'), we did not see any evidence of labeling of another dFB cell, whereas multiple PI cells were labeled.

It is not known whether electrical communication might be employed to promote sleep in other animal brains, although there is increasing evidence that gap junctions play an important role in the regulation of behavioral state and arousal in the mammalian brain (*Coulon and Landisman, 2017*). More generally, a role for gap junctions in sleep-promoting neurons also suggests a novel plasticity mechanism for regulating behavioral states. Whereas plasticity is typically viewed as a property of chemical synapses, the parallel electrical communication channel afforded by gap junctions could provide an alternate way of regulating sleep pressure, and thereby promoting sleep functions while ensuring that arousal thresholds are tightly linked to sleep need. In recent work, we have shown that transient electrical activation of the dFB neurons during fly brain development permanently impairs behavioral responsiveness in adult animals (*Ferguson et al., 2017*), further confirming the strong link between these neurons and the regulation of arousal in *Drosophila*. Different INX6 expression levels in the dFB may provide a mechanism for optimally linking behavioral responsiveness with sleep need. It will be interesting to see whether INX6 expression in the dFB correlates with the striking range of individual differences that we observed for sleep duration and behavioral responsiveness in wildtype flies (*Figure 1F*), and whether INX6 expression in the dFB might be co-regulated alongside other proteins that have been shown to control neuronal excitability in this arousal circuit (*Donlea et al., 2018*; *Pimentel et al., 2016*).

# Materials and methods

**Key resources table**

| Reagent type (species) or resource | Designation | Source or reference | Identifiers | Additional information |
|---|---|---|---|---|
| Genetic reagent (*D. melanogaster*) | R23E10-Gal4 | Bloomington | RRID:BDSC_49032 | |
| Genetic reagent (*D. melanogaster*) | C5-Gal4 | doi: 10.1002/ssscne.22284 | | Paul Shaw Lab |
| Genetic reagent (*D. melanogaster*) | 104y-Gal4 | Paul Shaw Lab | | |
| Genetic reagent (*D. melanogaster*) | UAS-CsChrimson | doi: 10.1038/nmeth.2836 | | provided by Vivek Jarayaman Lab |

*Continued on next page*

*Continued*

| Reagent type (species) or resource | Designation | Source or reference | Identifiers | Additional information |
|---|---|---|---|---|
| Genetic reagent (*D. melanogaster*) | UAS-2eGFP | Bloomington | RRID:BDSC_32186 | |
| Genetic reagent (*D. melanogaster*) | UAS-NaChBac | Bloomington | RRID:BDSC_9469 | |
| Genetic reagent (*D. melanogaster*) | UAS-syntaxin3-69 | Fly Base | FBal0092503 | |
| Genetic reagent (*D. melanogaster*) | UAS-shibireTS | Paul Shaw Lab | Gene ID: 45928 | |
| Genetic reagent (*D. melanogaster*) | tubpGAL80ts | Bloomington | RRID:BDSC_7108 | |
| Genetic reagent (*D. melanogaster*) | UAS-INX6 RNAi | VDRC | v8638 | Provided by the Chia-Lin Wu Lab |
| Antibody | Rabbit anti-INX6 | Provided by the Chia-Lin Wu Lab | | 1:1,000 |
| Antibody | Goat anti-mouse AlexaFluor488 | Invitrogen | Catalog # A-10680 | 1:200 |
| Antibody | Goat anti-rabbit AlexaFluor568 | Invitrogen | Catalog # A-11011 | 1:200 |
| Antibody | Goat anti-rabbit AlexaFluor647 | Invitrogen | Catalog # A-21244 | 1:200 |
| Antibody | Mouse anti-NC82 | DSHB | AB_2314866 | 1:10 |
| Antibody | Goat anti-rabbit AlexaFluor488 | Invitrogen | Catalog # A-11008 | 1:200 |
| Chemical compound, drug | Neurobiotin | Vector Labs | Cat. No: SP-1120 | |
| Software, algorithm | DART | bfklab | | http://www.bfklab.com/ |
| Software, algorithm | MATLAB code | This paper | 142faca | https://github.com/melvynyap/gap-junction-sleep-control (copy archived at https://github.com/elifesciences-publications/gap-junction-sleep-control) |
| Chemical compound, drug | All-trans retinal | SIGMA-Aldrich | SID 24899355 | |
| Chemical compound, drug | Vectashield | Vector Labs | Cat. No: H-1000 | |
| Chemical compound, drug | Streptavidin | Invitrogen | Catalog number: S32357 | 1:200 |

## Fly stocks and media

Flies were cultured on standard agar medium under a 12-hr day/night cycle. Flies used for optogenetics were placed on food containing 1 mM all-trans retinal (Sigma) for two days before experiments. All flies were outcrossed six generations to a w$^{2202}$ genetic background (isoCJ1; (*Yin et al., 1994*). The INX6-RNAi lines were a gift from Chia-Lin Wu. UAS-CsChrimson was a gift from Vivek Jayaraman. R23E10-Gal4, C5-Gal4, 104y-Gal4, and UAS-*shibire* were obtained from Bloomington. The INX6 RNAi is v8638, originally obtained from the VDRC (*Dietzl et al., 2007*). UAS-*syx*$^{3-69}$ was obtained from Bing Zhang.

## Behavioral experiments

One-day-old female virgin flies were briefly anesthetized on $CO_2$ for collection. At two days old, flies were transferred to 65 mm glass tubes (Trikinetics, Waltham, MA) sealed with food (containing retinal if necessary) at one end and cotton at the other and placed inside a 25°C incubator. A camera was positioned above for recording activity, and shaftless vibrating motors (Precision Microdrives, 312–101) were positioned underneath for stimulus delivery using DART (*Faville et al., 2015*). Four 700 mA 617 nm LED lights (Luxeon Star, Canada, SP-01-E4) were placed above the flies and

delivered irradiance of 0.03–0.1 mW/mm$^2$. These LEDs were controlled using a custom circuit board and Arduino with custom code.

After a day of acclimatization, fly activity was recorded over three days. Every hour, a stimulus was delivered consisting of five 0.2 s 2.4 g vibrations separated by 0.8 s. For optogenetic experiments, the red LEDs were switched on for the second and third day during ZT 0–12 periods to deliver either constant or 1 Hz (5 ms pulse width) light. For INX6 RNAi, sleep experiments were run for 3–4 days. All experiments were replicated at least three times. For UAS-Shibire[TS] and UAS-Syntaxin[3–69], experiments were run for 3 days and the temperature of the incubator was raised from 23°C to 31°C on days 2 or 3.

For acute R23E10-Gal4 activation experiments, the mechanical vibration was delivered to flies once every 15 min, alternating between periods with and without light exposure for six trials (three light on, three light off). The light was delivered for 1 min prior to and 3 min after the stimulus.

Sleep intensity was measured as the proportion of immobile (sleeping) flies that responded (at any level) to these stimuli. Flies were determined to have responded if they moved by a threshold of at least 3 mm (~3 body lengths) within the minute following the stimulus (*Faville et al., 2015*). To measure mean peak responsiveness, fly activity was averaged for two minutes prior to and the 15 min after each stimulus. This average activity was fitted with a single-inactivation exponential equation and the peak amplitude of activity following the stimulus was measured (*Faville et al., 2015*). To determine awake responsiveness, we included only flies that had moved within the 1 min prior to the stimulus (i.e. awake flies) in the analysis. Statistical analyses were performed in Prism 7 for Mac (GraphPad) and all tests were corrected for multiple comparisons.

For experiments in which we downregulated INX6 expression in adult flies (using R23E10-Gal4 > UAS-INX6 RNAi GAL80[TS]), flies were raised at 19°C. A four-day experiment used the same hourly stimulus protocol as above and consisted of 1 d at 19°C, 1 d at 31°C for GAL4 induction, followed by 2 d at 25°C. Data from both days following 31°C heating were combined and averaged.

## Immunochemistry

Fly heads were removed and brains dissected in cold PBS, before being fixed in 4% paraformaldehyde for 30 min. After washing with PBST (0.2% Triton-X and 0.01% sodium azide) and blocking with bovine serum albumin, brains were incubated with primary antibodies (R23E10-Gal4 > UAS-2eGFP: mouse anti-GFP 1:1,000, rabbit anti-INX6 1:1,000; CS: rabbit anti-INX6 1:1,000) overnight, and with secondary antibodies (R23E10-GAal4 > UAS-2eGFP: goat anti-mouse AlexaFluor488 1:200, goat anti-rabbit AlexFluor568 1:200; CS: AlexFluor687 1:200) for 1 d, then mounted using Vectashield. The INX6 primary antibody was a generous gift from Chia-Lin Wu. Samples were imaged on a spinning-disk confocal system (Marianas; 3I, Inc.) consisting of a Axio Observer Z1 (Carl Zeiss) equipped with a CSU-W1 spinning-disk head (Yokogawa Corporation of America), an ORCA-Flash4.0 v2 sCMOS camera (Hamamatsu Photonics), and 20 × 0.8 NA PlanApo and 100 × 1.4 NA PlanApo objectives. Image acquisition was performed using SlideBook 6.0 (3I, Inc). To quantify the degree of colocalization between INX6 and R23E10-Gal4 dFB neurons, we used ImarisColoc (Imaris, Bitplane Inc). INX6 and GFP channels were selected using automatic threshold detection from pixel intensity histograms. The number of voxels in these channels that colocalized across Z-planes inside of the dFB region of interest (ROI) was then determined. To determine INX6 RNAi efficacy, we compared INX6 expression in the dFB in control brains (Elav-Gal4/+) to brains expressing the RNAi (Elav-Gal4 >UAS-INX6 RNAi) using an INX6 antibody. Brains were incubated with primary antibody (rabbit anti-INX6 1:1,000, mouse anti-NC82 1:10) overnight, and with secondary antibody (goat anti-rabbit AlexaFluor488 1:200, goat anti-rabbit AlexFluor647 1:200) for a day. A ROI was drawn around the dFB using the NC82 staining. Mean INX6 intensity within this ROI was normalized to mean NC82 intensity. Normalized dFB immunoreactivity in control brains was compared to Elav-Gal4 >UAS-INX6 RNAi brains using a t-test.

## Quantitative PCR

A quantitative reverse transcriptase PCR (qRT-PCR) assay was used to conclude whether INX6 knockdown was achieved relative to that of the housekeeping gene Act88F, which was determined to be stably expressed across all experimental conditions. Females of 3–5 d old were collected by $CO_2$ anesthesia, snap frozen, and stored at −80°C. Six pools of five fly heads (30 heads total) were placed

into a 1.5 ml Eppendorf tube. Total RNA was purified using TRIzol according to the manufacturer's protocols (Invitrogen, Carlsbad, California), immediately after dissection. Total RNA was treated with DNase (Sigma-Aldrich, St. Louis, MO) to eliminate contaminant genomic DNA. Approximately 0.5 µg of total RNA was reverse transcribed using random primers (Invitrogen) and reverse transcriptase (Invitrogen) according to the manufacturer's protocols. Gene expression was estimated with two technical replicates using a standard quantitative PCR (qPCR) assay (*McMeniman et al., 2009*). Each qPCR mixture contained 12.5 µL of 2X SYBR premix (Invitrogen), 1 µL of forward primer, 1 µL of reverse primer, 100 ng of DNA, and $H_2O$ to a final volume of 25 µL. The expression of two genes was estimated relative to Act88F using the CT (where CT is the threshold cycle) method (*Pfaffl, 2001*). Averages of expression were compared using Student's t test (in SPSS software). The primers that were used were actinF:ATCGAGCACGGCATCATCAC, actinR:CACGCGCAGCTCG TTGTA, inx6(v8638)F:GAACGGCATGCCCAAGTC, and inx6(v8638)R:ACCAGTCGCCGTATCCAG.

## Electrophysiological recordings

Methods for performing in vivo open brain recordings (*Figure 3A*) were described elsewhere (*Yap et al., 2017*). Briefly, to prepare for brain electrical recordings, we secured a 3–7-d-old fly on a custom fly plate (*Maimon et al., 2010*) (*Figure 3A*). The bath chamber of the fly plate was filled with oxygenated (bubbled in 95% $O_2$, 5% $CO_2$) extracellular fluid (ECF) containing (in mM): 103 NaCl, 10.5 trehalose, 10 glucose, 26 NaHCO$_3$, 5 $C_6H_{15}NO_6S$, 5 MgCl$_2$ (hexa-hydrate), 2 sucrose, 3 KCl, 1.5 CaCl (dihydrate), and 1 NaH$_2$PO$_4$. To enable access to the brain, the cuticle was partially removed using forceps, with the perineural sheath removed either mechanically with forceps or chemically using protease (0.5% collagenase type IV). The fly in this preparation was positioned on an air-supported ball and the brain was continuously superfused with oxygenated ECF.

Whole-cell patch clamp recordings were obtained from the cell bodies of the R23E10 neurons using a fixed-stage upright fluorescence microscope (Olympus BX51WI, U-RFL-T, Olympus, Berlin, Germany) with a 40x water-immersion objective. To achieve high-contrast visuals, an infrared LED (Osram SFH 4232) coupled with an infrared camera (DAGE-MTI IR-1000) was used, whereas visualization of the GFP-labeled neurons was achieved using a mercury short-arc lamp (HBO 103 W/2). Electrode positioning was controlled by a motorized micromanipulator system (Sutter MP-285).

Borosilicate glass capillaries (Harvard Instruments GC150F-25) were used, pulled (Sutter P-97 micropipette puller) to a tip size of 8–12 MΩ, and subsequently filled with an internal solution containing (in mM): 140 potassium aspartate, 10 HEPES, 1 KCl, 4 MgATP, 0.5 Na$_3$GTP, 1 EGTA, and 0.05 Alexa fluor 568, pH 7.3, adjusted to 265 mOsm. Voltage signals were acquired in current-clamp mode, with a CV-7B headstage (Molecular Devices) and a Multiclamp 700B amplifier (Molecular Devices), low-pass filtered at 10 kHz (Bessel), and digitized at 10 kHz using an Axon Digidata 1440A Digitizer controlled by the MultiClamp 700B Commander Software and AxoGraph X 1.4.4 (Axon Instrument) on a computer running Windows XP. In some cells, it was necessary to inject a small constant hyperpolarizing current (15–40 pA) in order to stabilize the resting membrane potential to between −30 and −50 mV.

LFP recordings were performed simultaneously with whole-cell recordings using same-sized micropipettes filled with ECF instead of internal solution. LFP signals were acquired with a FET electrode, amplified and filtered (low: 0.1 Hz, high: 1 kHz) (A-M Systems Model 1700), digitized (Axon Digidata 1440A Digitizer) and sampled at 10 kHz using the data acquisition software AxoGraph X 1.4.4.

## Electrophysiology and optogenetic stimulation

Photostimulation of CsChrimson-expressing neurons was achieved by using an ultra-bright red LED (617 nm Luxeon Rebel LED, Luxeon Star LEDs, Ontario, Canada) directed to the opened section of the fly head, producing 0.1–0.2 mW/mm$^2$ at a distance of 4–5 cm with the aid of concentrator optics (Polymer Optics 6° 15 mm Circular Beam Optic, Luxeon Star LEDs). To prevent overheating of the fly and the immediate environment, the LED was mounted onto a sink pad (SinkPAD-II 20 mm Star Base) that was attached to a small heat sink. The temperature of the solution bath was also kept constant within the range of 22°C to 23°C by using a thermistor and an in-line heater/cooler (Warner Instruments Model SC-20), both driven by a temperature controller (Warner Instruments Model CL-100). Light exposure was triggered after 1 min of baseline recording and lasted for 2 min. Light was delivered either in a continuous or in a pulsatile fashion, the latter of which consists of a 1 Hz train of

5 ms optical pulses. Timing of the light switch was controlled by AxoGraph, which also measured the timing of light exposure from a basic photodiode.

## Dye labeling following electrophysiology recordings

Neurons in the dFB were injected with internal solution supplemented with 0.5% neurobiotin (Vector Labs TM), which has been shown to pass through gap junctions in invertebrates effectively (*Fan et al., 2005*). Whole-cell configuration was achieved for the purpose of passive flow of dye into the cytoplasm. Dye injection was aided by iontophoresis, with the delivery of depolarizing current pulses (200 pA, 500 ms, 1 Hz, 50% duty cycle) delivered over 5 min, followed by hyperpolarizing current pulses ($-200$ pA). The preparation was left for a further 40 min to allow the dye to diffuse into the cell passively. After withdrawing the micropipette, the preparation was left untouched for about 1 h to allow sufficient time for dye to diffuse within and across cells.

The fly brain was dissected and fixed at room temperature for 20 min in 4% paraformaldehyde in 0.1M phosphate buffer (pH 7.0) then washed $3 \times 10$ min in PBST. The brain was incubated for 24–48 hr at 4°C with streptavidin conjugated with Alexa Fluor 647, in 5% bovine serum albumin (BSA) (1:200). Brains were rinsed in PBST ($3 \times 10$ mins) and were mounted on glass slides in Vecta Shield mounting medium. All confocal images were obtained using an inverted spinning disk microscope (Yokogawa W1). All confocal images obtained were analyzed and processed in Fiji/Imaris. Deconvolution of images acquired using the Yokogawa W1 spinning disc confocal microscope was performed with Huygens Professional Plus Deconvolution software (Scientific Volume Imaging, Hilversum, The Netherlands). We used a theoretical point spread function (PSF) that was obtained by the parameters of image acquisition. For the deconvolution, we used a total image change threshold of 0.01, with single block processing on, a maximum iteration value of 40 and a signal to noise ratio (SNR) of 20.

## Gap-junction blocker delivery

For the gap junction blocker experiments, carbenoxolone (CBX, 1 mM) (*Cao and Nitabach, 2008*) was dissolved into ECF, which was delivered to the fly brain by either perfusing the bath chamber with it or locally injecting it into the dFB region via a micropipette attached to a micromanipulator. For local delivery, constant air pressure was applied to push the CBX solution out of the micropipette, with the flow of solution into the dFB confirmed using a fluorescent dye (Alexa Fluor 568) that was added to the CBX solution. LFP recordings were obtained prior to bath perfusion of CBX, which served as baseline for comparison with the CBX effect. As an additional control for the locally delivered CBX experiments, ECF with no added CBX (vehicle) was injected to observe the effect of applying physical pressure into the central brain.

## Electrophysiological analyses

All analyses were performed offline using custom scripts in MATLAB (2014a) (*Yap, 2018*) (https://github.com/melvynyap/gap-junction-sleep-control, 142faca; copy archived at https://github.com/elifesciences-publications/gap-junction-sleep-control). Cells were identified as burst-spiking cells when an action potential failed to return to resting membrane potential prior to reaching the next peak. These cells, when hyperpolarized with a constant current, also failed to produce single spiking events. Cells observed to have at least one single spiking event in addition to the burst spiking were identified as having a pattern involving both single and burst-spiking events. Non-spiking cells were identified as such when the cell failed to produce any action potentials despite the injection of depolarizing current in steps of +10 pA up to +150 pA.

We use the change in membrane potential ($\Delta V_m$) as the measure of cell response to photostimulation. For the constant light stimulus condition, the mean membrane potential for the pre-stimulation period in each cell was obtained over the length of baseline, which was 1 min. Mean membrane potential during stimulation was calculated over the first 1 min of photostimulation, whereas for post-stimulation membrane potential, a mean was calculated over the last 1 min of recording (1 min after the stimulation ended). The mean membrane potential for each cell for the 1 Hz pulse photostimulation was calculated by first producing a composite membrane potential that represents the average of all 120 pulses, followed by obtaining the average membrane potential for the first 100

ms of this composite data. From these composite membrane potentials, it was visually evident that the 1 Hz light pulse appeared to be having an effect on the spike timing in some cells.

Owing to the unipolar morphology of the recorded cells, the true value of the cells' resting membrane potentials were probably lower and less variable than those recorded from the cell bodies (*Gouwens and Wilson, 2009*). Therefore, to better represent the effect of photostimulation of the CsChrimson-expressing cells, the data were displayed as $\Delta V_m$ instead of absolute values. Cells with various single-spike and burst-spiking characteristics were found to exhibit the same membrane potential response, and were therefore combined for the purpose of statistical analyses. In a subset of cells, we observed a volley of evoked spikes in response to the constant light stimulus that lasted very briefly. Interspike intervals were obtained from these spikes by measuring the time between the first and second peak.

Raw LFP signals were transformed into wavelets using the Morlet wavelet transformation function ft_specest_wavelet in the Fieldtrip MATLAB toolbox, with wavelet width value set at 30 and 3 standard deviations. To generate the averaged time-frequency spectrograms, wavelets were normalized to the mean wavelet values of the pre-stimulus segment (baseline) for each fly prior to averaging across all flies. Wavelets differ in magnitudes across fly recordings, and were therefore normalized for each fly prior to averaging. For statistical analyses of 1–15 Hz LFP power, wavelets were also normalized to the mean wavelet values of baseline but were done within the respective frequency bins of 0.1 Hz, producing wavelet ratios (to baseline). The mean wavelet ratios during the stimulation period were calculated for each fly and then averaged across flies. Baseline was subsequently zeroed so that any positive values obtained during the stimulation period denotes an increase in LFP power. Peak amplitude from raw LFP signal represents the LFP response to the pulsative light stimulation and as such was defined as the maximum field potential deviation from zero in the 50 ms after the onset of the 1 Hz 5 ms light pulse. Peak amplitude was averaged across all 120 trials per fly prior to averaging across flies.

## Statistical analyses

All statistical analyses were performed using Prism 7 for Windows (GraphPad). Where dataset failed the Shapiro-Wilk normality test ($p < 0.05$), non-parametric tests were used. Friedman test with Dunn's post hoc multiple comparisons test were used to compare the membrane potential averages during the stimulation and post-stimulation period to pre-stimulation baseline period. The Wilcoxon signed rank test was used to test for significant difference between the LFP power ratio and a baseline of zero during the stimulation period. The Mann-Whitney test was used to compare the peak LFP amplitudes between the two fly strains. Paired t-tests were used to compare the peak LFP amplitudes between the CBX and baseline conditions. All membrane potential data in the figures are presented as median and range, whereas the LFP data presented in figures are shown as means ± SEM. All tests for significance were two-tailed and confidence levels were set at $\alpha = 0.05$.

## Acknowledgements

We thank Leonie Kirszenblat for help and comments on the manuscript. We thank Eleni Notaras for help with behavioral experiments. We also thank Chia-Lin Wu for the INX6 antibody. This work was supported by an NIH grant RO1 NS076980-01 to PJS and BVS, and by an NHMRC grant GNT1065713 to BVS. The authors declare no conflicts of interest.

## Additional information

### Funding

| Funder | Grant reference number | Author |
|---|---|---|
| National Institutes of Health | R01 NS076980 | Melvyn HW Yap |
| | | Paul Shaw |
| | | Bruno van Swinderen |

National Health and Medical      GNT1065713          Michael Troup
Research Council                                     Chelsie Rohrscheib
                                                     Aoife Larkin
                                                     Bruno van Swinderen

The funders had no role in study design, data collection and interpretation, or the decision to submit the work for publication.

## Author contributions

Michael Troup, Conceptualization, Resources, Data curation, Software, Formal analysis, Supervision, Validation, Investigation, Visualization, Methodology, Writing—original draft, Writing—review and editing; Melvyn HW Yap, Conceptualization, Resources, Data curation, Software, Formal analysis, Validation, Investigation, Visualization, Methodology, Writing—original draft; Chelsie Rohrscheib, Resources, Formal analysis, Investigation, Visualization, Methodology; Martyna J Grabowska, Deniz Ertekin, Resources, Software, Formal analysis, Investigation, Visualization, Methodology; Roshini Randeniya, Formal analysis, Visualization, Methodology; Benjamin Kottler, Conceptualization, Resources; Aoife Larkin, Data curation, Formal analysis, Visualization, Methodology; Kelly Munro, Formal analysis, Investigation, Visualization; Paul J Shaw, Conceptualization, Resources, Supervision, Funding acquisition, Methodology, Writing—original draft, Project administration, Writing—review and editing; Bruno van Swinderen, Conceptualization, Resources, Data curation, Formal analysis, Supervision, Funding acquisition, Validation, Investigation, Visualization, Methodology, Writing—original draft, Project administration, Writing—review and editing

## Author ORCIDs

Martyna J Grabowska (iD) http://orcid.org/0000-0002-1727-7714
Roshini Randeniya (iD) http://orcid.org/0000-0002-1340-750X
Benjamin Kottler (iD) http://orcid.org/0000-0002-4551-5791
Bruno van Swinderen (iD) http://orcid.org/0000-0001-6552-7418

## Decision letter and Author response

Decision letter https://doi.org/10.7554/eLife.37105.024
Author response https://doi.org/10.7554/eLife.37105.025

# Additional files

## Supplementary files

• Transparent reporting form
DOI: https://doi.org/10.7554/eLife.37105.021

## Data availability

All data generated or analysed during this study are included in the manuscript and supporting files.

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
