## [Decision Letter]

[Editors’ note: this article was originally rejected after discussions between the reviewers, but the authors were invited to resubmit after an appeal against the decision.]

Thank you for submitting your work entitled "Acute control of the sleep switch in *Drosophila* reveals a role for gap junctions in regulating behavioral responsiveness" for consideration by *eLife*. Your article has been reviewed by 3 peer reviewers, including Mani Ramaswami as the Reviewing Editor and Reviewer #1, and the evaluation has been overseen by a Senior Editor.

Our decision has been reached after consultation between the reviewers. Based on these discussions and the individual reviews below, we regret to inform you that your work will not be considered further for publication in *eLife*.

All the reviewers observed the intrinsic interest of the questions being addressed: the control of arousal threshold, potentially independently of sleep; and underlying cellular mechanisms. They also were universally appreciative of the impressive range of techniques employed in this study. However, an overriding concern was the need for a substantial number additional experiments to establish the main premise of the paper that a mechanism for reducing arousal threshold is deployed prior to one involved in sleep induction. Moreover, there also remained some questions on the role of *innexin6* function, in particular on its role in baseline sleep and in neurons other that dFB neurons.

Reviewer #1:

Troup and coauthors analyse mechanisms by which R23E10 marked fan-shaped body neuron control sensory responsiveness and sleep states. Previous work has shown that these neurons mediate and drive entry into the sleep state in response to sleep need, but has not described whether they influence sensory responsiveness that must decrease prior to entry into sleep. The authors show that activation of dFB neurons also reduces sensory responsiveness and,by studying this in actively moving animals, present evidence to show that this effect is prior to and independent of entry into the sleep state. Building on this through a variety of technically impressive experiments, the main conclusion from this work is that these dFB neurons act to reduce sensory responsiveness prior to and independent of their role in promoting the sleep state. This independent function is proposed to be mediated by *innexin6* gap junctions expressed in these cells that allow coherent activity among connected dFB neurons that is reflected in stimulation-evoked LFP potentials that oscillate in the 1-15 Hz power range.

1) The role for *innexin6* in coupling dFB neurons is interesting and solid and the experiments seem well done. Without experiments that hint at differences in downstream mechanisms, one is left with the impression that differences in the *innexin6* requirement for sensory responsiveness vs sleep may in some way reflect different thresholds of dFB activity, rather than two distinctive downstream pathways as proposed by the authors. Perhaps the paper could be summarised and considered as a Short Report?

2) Figure 1 sensory responsiveness and sleep are only correlated at night in wild-type flies, but this correlation is seen in the day as well following dFB stimulation. Could this be an artefact of the level of sleep? Another way of interpreting Figure 1F and H is to state that correlations emerge when sleep duration is more than 30 mins/hour. If this is a major point, it needs to be better justified. If minor (as it appears), then it either should not be made or should be duly qualified.

3) While Figure 2 shows that dFB activation results in reduced responsiveness in active flies, it does not clarify whether this is a threshold effect – ie. FB activation and resulting GABA release results in reduced responsiveness at low levels of inhibition, but requires higher levels of inhibition of the same downstream substrates for sleep. Perhaps the levels of GABA release required are different in day and night?

4) The experiments themselves seem nicely done, but some of the figures make points, which though important to establish, are not highly instructive in themselves. E.g. Figure 3 shows that R23E10 neurons are different in their intrinsic properties (though the significance of this is unclear) and that light driven ChR does indeed result in neuronal firing. Figure 4 shows that *innexin6* is expressed in at dFB neurons. The authors may consider integrating these points in summary form with one of the other figures and show more comprehensive images in the figure supplements.

Reviewer #2:

This is an interesting paper on fly sleep from the van Swinderen and Shaw labs. The paper follows up recent results from van Swinderen's lab, which reported local field potential oscillations in the fly ellipsoid body/central complex, and these oscillations vaguely resemble the famous thalamo-cortical oscillations characteristic of mammalian sleep. These previously observed oscillations were linked to arousal threshold and fly sleep. Arousal threshold is a fundamental property of sleep in all organisms and is therefore important. The relationship or arousal threshold to other aspects of sleep is also of interest. These authors focus on the dorsal fan shaped body (dFB) an important sleep-promoting sub-structure of the central complex.

There are 3 central results: (1) Optogenetic activation of the dFB increases arousal threshold as well as other sleep parameters. (2) The effect on arousal threshold is faster or more complete than the effect on other sleep parameters, suggesting that the dFB has a more primary or more prominent role in affecting arousal threshold than on other aspects of sleep like sleep duration. (3) Gap Junctions and in particular *innexin6* is relevant to this arousal threshold function of the dFB.

1) The R23E10-Gal4 driver is not so clean and has some VNC expression. Even if the authors dispute this statement, it is prudent to use at least two drivers with dFB expression and show that both give comparable results. For example: R84C10-GAL4 is a strong and clean dFB-GAL4 driver.

2) It is important to use inhibition as well as activation to back up the claim that the dFB plays an important role in arousal threshold. (Sufficiency does not ensure necessity, and we should now learn which it is.) The Discussion section is written as if this had been done. This request would have been unfair a couple of years ago, but new optogenetic tools have come on-line with widespread use and endorsement. If inhibition fails to have an effect, I would not recommend rejection of the paper, but I think the authors should find out.

3) In Figure 7, the authors showed the behavioral responsiveness to stimulus when INX6 knockdown in non-activated R23E10 neurons, and it is comparable to control flies. Does this mean that gap junctions in dFB neurons play no role in arousal regulation under basal conditions? If so, what do the authors think about this?

4) The knockdown experiments should be done with tubpGAL80ts to eliminate developmental effects. And why were the MB lobes were not identified with INX6 staining? In Wu et al. (2011), they reported that INX6 was expressed in the MB and EB as well as the FB.

5) Why is daytime baselines sleep so different in Figure 7A than in Figure 5A? This makes me doubt whether the important result in Figure 5A is correct, but perhaps I am missing something. This difference is also true for nighttime sleep, which showed an effect in Figure 5A but no apparent effect in Figure 7A. I understand that in Figure 7A they co-expressed UAS-CsChrimson with INX6 RNAi but CsChrimson should not affect INX6 RNAi-induced less sleep phenotype on the baseline day unless there is a subtle RNAi baseline effect. This should then be tested with GFP RNAi or something like this.

6) I am not convinced the order of events (first arousal threshold, then sleep) is as the authors suggest. The former can be assayed instantly whereas the latter requires some time to qualify as sleep.

7) Lots of control data is missing for the behavior experiments. For example, GAL4 and UAS control are needed in Figure 2 and Figure 7.

8) Error bars are needed in Figure 2—figure supplement 2A and Figure 2—figure supplement 2E. And why is the behavioral response in Figure 2—figure supplement 2G on the baseline day is so different compared to Figure 2—figure supplement 2C on the baseline day? They should be the same genotype and condition. And what is the explanation why for the 10Hz light pulse activation of R23E10 has *no* effect on sleep duration?

Reviewer #3:

In this manuscript, the authors attempt to define the cellular and molecular control of behavioral responsiveness in flies. The work is well-presented and technically very impressive. However, I have major experimental and conceptual concerns. Most importantly, the novelty of the manuscript is in trying to explain how behavioral responsiveness changes to permit sleep, with the key point being that sleep pressure must alter responsiveness prior to sleep onset itself. This is an intriguing idea but the evidence is lacking. The major concerns are as follows:

1) Figure 2 is essential in making the case that dFB activation reduces behavioral responsiveness separately from (and perhaps prior to) sleep. However, it is perplexing that responses are "abolished" (subsection section “Optogenetic activation of the sleep switch decreases behavioral responsiveness”) in wake flies with dFB activation, while in sleeping flies, the response is only diminished. Is this a function of needing to normalize to wake activity (Figure 2E)? If so, is wake activity (mean speed) affected by dFB activation? How about movement in response to a strong stimulus? This raises the more general possibility that locomotor function is simply different with dFB activation, as opposed to an actual change in responsiveness to the stimulus at a sensory level. Can the authors address this?

2) Also in Figure 2, I am confused that the earlier work (Figure 2C and D, excluding sleep intensity) do not distinguish between wake and sleeping flies. Why not only analyze sleeping flies until shifting focus to wake flies in Figure 2E? If analysis of wake flies requires normalizing the baseline activity (Figure 2E right panel), then shouldn't the authors need to account for that in analyzing mean peak responsiveness (Figure 2D, for example)?

3) Figure 2F is particularly critical, and the authors set it up as if it will answer something about whether the change in responsiveness or sleep comes first (subsection section “Optogenetic activation of the sleep switch decreases behavioral responsiveness” "even in awake flies"). However, it appears to again mix/combine sleeping and awake flies for analysis, which is a major confound. In other words, what if most of these flies are rapidly put to sleep with the 1 minute dFB activation prior to stimulus? Then the figure simply shows that inducing sleep with dFB changes responsiveness, which we already know (from this work, but also from Donlea et al. (2011) and Donlea, Pimentel and Miesenbock (2014) – it is important to note that these published works have already demonstrated a change in arousability with dFB activation, even if not with such detail as shown here).

4) The authors make a point of 1Hz and 10Hz optogenetic activation of dFB both changing responsiveness, but only 1Hz inducing sleep, therefore dissociating the two. However, these data are not convincing. In Figure 2—figure supplement 2G, why is the response so poor at baseline compared to C in this same figure? And why suddenly so much noise? Moreover, the change in responsiveness (H) appears similar +/- ATR: the baselines appear different and the 10Hz light causes a similar decrease in both (same issue in D of this figure). In other words, I am not convinced that there is a change in responsiveness, and therefore the ability to dissociate sleep time from arousal is hindered. The sleep intensity data is more convincing, but this is a comparatively less precise measure.

5) In Figure 7, there are concerns. First, in contrast to Figure 5A, there is no longer an effect of INX6 RNAi expressed using R23E10-GAL4 (Figure 7A, left). Is there dilution of UAS-driven expression by co-expressing two UAS? Is INX6 knockdown even effective here? If we assume so, and if INX6 is mostly important in the context of dFB activation, then the authors should test these same measures following sleep deprivation. That would be most convincing with regard to a role for INX6 in modulating dFB driven sleep intensity. Overall, it would add a lot to this work for the authors to perform many of the experiments in Figure 2 after sleep deprivation, which is more physiologically relevant and should induce changes in behavioral responsiveness.

---

## [Author Response]

[Editors’ note: the author responses to the first round of peer review follow.]Reviewer #1:[…]1) The role for innexin6 in coupling dFB neurons is interesting and solid and the experiments seem well done. Without experiments that hint at differences in downstream mechanisms, one is left with the impression that differences in the innexin6 requirement for sensory responsiveness vs sleep may in some way reflect different thresholds of dFB activity, rather than two distinctive downstream pathways as proposed by the authors. Perhaps the paper could be summarised and considered as a Short Report?

It is possible that different thresholds of dFB activity are involved for sleep and responsiveness. However, we get similar behavioral and electrophysiological results when we stimulate for 5ms at 1Hz as for when stimulate continuously with red light. As the 1Hz regime involves 99.5% less red light, it seems unlikely that this is a threshold effect. Rather, thresholds could be determined in a different way. Indeed, our 10Hz stimulation data suggests a likely frequency effect: 10Hz suppressed responsiveness but did not induce sleep. This already suggests a dissociation between these processes, which is a key theme throughout our paper. Dissecting this in the frequency domain will be an interesting future research direction, however this is not what this paper is about. To prevent any confusion, we are inclined to remove the 10Hz data from our paper (see replies to other reviewers, below). We believe the 1Hz regime is sufficient to further convince that our activation effects are real.

2) Figure 1 sensory responsiveness and sleep are only correlated at night in wild-type flies, but this correlation is seen in the day as well following dFB stimulation. Could this be an artefact of the level of sleep? Another way of interpreting Figure 1F and H is to state that correlations emerge when sleep duration is more than 30 mins/hour. If this is a major point, it needs to be better justified. If minor (as it appears), then it either should not be made or should be duly qualified.

It is minor, and we have altered the text to qualify this (e.g., Results subsection “Correlating sleep duration and behavioral responsiveness”). The point is simply sleep and responsiveness can be separated (day and night) or made more tightly linked (by dFB activation). However, it is difficult to disentangle responsiveness and sleep duration with chronic manipulations, and we now make this point clear (in the aforementioned subsection). This first result is important to confirm the tight correlation between these processes.

3) While Figure 2 shows that dFB activation results in reduced responsiveness in active flies, it does not clarify whether this is a threshold effect – ie. FB activation and resulting GABA release results in reduced responsiveness at low levels of inhibition, but requires higher levels of inhibition of the same downstream substrates for sleep. Perhaps the levels of GABA release required are different in day and night?

It is unlikely to be a threshold effect, because flies stop responding within a minute of dFB activation (Figure 2F), but then the same stimulus regime if left on will lead to increased sleep duration and intensity (Figure 2B and C). Of course, this could be a cumulative effect, but then that would be exactly the sequence of events we might predict from increased arousal thresholds preceding deeper sleep induction. We now provide some more support for this sequence of events in sleep intensity data for R23E10/Chrimson flies (new Figure 2—figure supplement 1D), which is separated in 5min bins. If further exposed to red light, these flies will sleep more deeply, so there appears to be a sequential effect here of decreased behavioral responsiveness while awake followed by increased sleep intensity.

While the GABA-related questions are thought provoking, we are not investigating GABA pathways in this paper. We are showing that removing INX6 from the sleep switch impairs its ability to block behavioral responsiveness upon activation. GABA could still play a role, but our story is about INX6, and the effects are clear.

We decided to nevertheless probe potential downstream synaptic effects, by expressing the temperature sensitive synaptic effectors UAS-Shibire^TS^ and UAS-Syntaxin^3-69^ in R23E10 neurons. These manipulations should have opposite effects: UAS-shibire impairs synaptic release in neurons upon exposure to elevated temperatures (31°C), whereas syx^3-69^ increases synaptic release at elevated temperatures. In neither case did we see convincing evidence of altered behavioral responsiveness (or sleep), compared to genetic controls (Figure 4—figure supplement 1). This is in striking contrast with the evidence of effects with INX6 manipulations (see Author response image 1 below). While it remains possible that these synaptic manipulations might not influence some of the downstream synaptic effects likely to be involved here (e.g., neuropeptides or GABA), and there might also be a confounding effect of temperature, we feel these negative results highlight the importance of the electrical activation and INX6 effects we have discovered, which is what our work is focussed on. We have included these synaptic manipulations as new supplement Figure 4—figure supplement 1 in the revised manuscript, as it also provides a good rationale for why we proceeded to investigate electrical communication channels instead.

4) The experiments themselves seem nicely done, but some of the figures make points, which though important to establish, are not highly instructive in themselves. E.g. Figure 3 shows that R23E10 neurons are different in their intrinsic properties (though the significance of this is unclear) and that light driven ChR does indeed result in neuronal firing. Figure 4 shows that innexin6 is expressed in at dFB neurons. The authors may consider integrating these points in summary form with one of the other figures and show more comprehensive images in the figure supplements.

Showing that INX6 is expressed in the dFB is critical for the paper, and we went through quite a bit of work to quantify the overlap between INX6 labeling and R23E10 expression (in the Methods). We are not aware of any whole-cell patch electrophysiology showing sleep switch cell behavior during optogenetic induction, so feel those are valuable figures. Showing the intrinsic properties of these neurons aligns our findings with what others have shown, and this is often a valuable starting point for any electrophysiological study. We can however reconsider how to present our figures to move more effectively to the questions of greatest interest.

Reviewer #2:[…]1) The R23E10-Gal4 driver is not so clean and has some VNC expression. Even if the authors dispute this statement, it is prudent to use at least two drivers with dFB expression and show that both give comparable results. For example: R84C10-GAL4 is a strong and clean dFB-GAL4 driver.

It is true that R23E10 has some VNC expression, but the evidence of multiple studies (including our recent paper, Yap et al., 2017) point to the dFB as relevant for these sleep effects, with the R23E10 strain being accepted by the field as a sleep switch Gal4. We have tested R84C10-Gal4 as suggested, and, interestingly, it is not sleep promoting. So, we did not pursue that Gal4 in this paper. This suggests cellular heterogeneity in the dFB, which is the subject of a future paper.

2) It is important to use inhibition as well as activation to back up the claim that the dFB plays an important role in arousal threshold. (Sufficiency does not ensure necessity, and we should now learn which it is.) The Discussion section is written as if this had been done. This request would have been unfair a couple of years ago, but new optogenetic tools have come on-line with widespread use and endorsement. If inhibition fails to have an effect, I would not recommend rejection of the paper, but I think the authors should find out.

We acquired the inhibition reagents (GtACR) but have had limited success in getting them to work reliably, at least in the context of R23E10 expression. On first blush it appears that expressing GtACR in the dFB increases responsiveness as predicted (see exemplary traces and summary stats in Author response image 1). However, this effect seems independent of green light activation, suggesting a leakiness due to ambient light. Then, it also seemed like acute dFB inhibition suppressed behavioral responsiveness (Author response image 1), similar to what we saw with acute activation (Figure 2F in the paper). However, the non-ATR fed controls show a similar effect, if not stronger suppression (see Author response image 1). There are multiple, opposing ways to interpret this experiment. Perhaps the green light (to activate the inhibitory channel) causes the control flies to freeze, and this is suppressed when the dFB is acutely inhibited. This would be consistent with our view of a role for the dFB in regulating behavioral responsiveness. Or, it could be the exact opposite: acute dFB inhibition might be suppressing responsiveness, and controls might be showing this too because the effect is leaky even without ATR feeding. Or, the fact that this reagent might already respond to ambient light could be an issue: this could explain the greater overall responsiveness in ATR-fed flies, even outside of the acute green light exposure, compared to non-ATR controls. We see similar results with prolonged light exposure (Author response image 1), and, oddly, sleep also increases in ATR-fed flies regardless of light exposure (Author response image 1). Clearly there is a leakiness issue here, so these experiments are difficult to interpret. Our worry is that including these manipulations in the paper (we have repeated and confirmed this) will not help explain much without a lot more optimisation. Including these results will just raise more questions and will break the flow of our activation-centred narrative. Optimising this paradigm to understand what dFB inhibition is actually doing, and to exclude a large number of alternative explanations, will require almost a paper of its own, with associated electrophysiology. Our suspicion is the acute green light induces freezing behavior in some flies (which is already interesting), that this is suppressed when the dFB is inhibited (which could be really interesting), but that GtACR is responding already to ambient light (which is troubling).

However, this is quite tangential to our current story, so I would rather keep this out of the revised manuscript.

**Author response image 1. respfig1:** Acute inhibition of the dFB. (**A**) Example average activity trace (speed ± SEM) of ATR-fed R23E10/GtACR-3M flies responding to stimuli 15 minutes apart (gray dashed lines); 5min GtACR activation (green shading), with 1min prior to the stimulus, is alternated with trials without green light. (**B**) Same acute experiment as in A, for non-ATR fed controls. (**C**) Average responsiveness ( ± SEM) in same flies as in A&B, for a 12-hour experiment. (**D**) Average sleep duration ( ± SEM) for same flies as in A&B, for a 12-hour experiment. N=16 flies for all..

3) In Figure 7, the authors showed the behavioral responsiveness to stimulus when INX6 knockdown in non-activated R23E10 neurons, and it is comparable to control flies. Does this mean that gap junctions in dFB neurons play no role in arousal regulation under basal conditions? If so, what do the authors think about this?

Mean peak responsiveness (Figure 7D) is the odd one out here, for baseline; both sleep duration and more importantly, sleep intensity, show the expected difference at baseline. Peak responsiveness refers to fly speed immediately after the mechanical stimulus. It is possible that leaky Chrimson expression in ambient light already attenuated the baseline peak responsiveness. We now acknowledge this one baseline inconsistency in our Results section and suggest the likely explanation. Importantly, the effect during acute dFB activation is really clear: INX6 knockdown in the dFB completely blocks the optogenetic effect on behavioral responsiveness.

4) The knockdown experiments should be done with tubpGAL80ts to eliminate developmental effects. And why were the MB lobes were not identified with INX6 staining? In Wu et al. (2011), they reported that INX6 was expressed in the MB and EB as well as the FB.

It isn’t clear to us why we need to show MB staining here? The antibody is working. We could focus on the MB, and turn up the gain. It’s in there. But the dFB is brighter, and this is a dFB study. For tub-Gal80, we have done those experiments, and now include them in the paper (Figure 5—figure supplement2). While effects on sleep duration are not significant, it is clear that daytime responsiveness and sleep intensity are showing the same significant effects if INX6 is downregulated in adults (pink) as for as the chronic manipulation (red, repeated here). This argues against any developmental effects. The lack of significant effects on sleep duration also further solidifies our view that INX6 is primarily affecting behavioral responsiveness levels.

5) Why is daytime baselines sleep so different in Figure 7A than in Figure 5A? This makes me doubt whether the important result in Figure 5A is correct, but perhaps I am missing something. This difference is also true for nighttime sleep, which showed an effect in Figure 5A but no apparent effect in Figure 7A. I understand that in Figure 7A they co-expressed UAS-CsChrimson with INX6 RNAi but CsChrimson should not affect INX6 RNAi-induced less sleep phenotype on the baseline day unless there is a subtle RNAi baseline effect. This should then be tested with GFP RNAi or something like this.

Both are around 20min per hour, for daytime sleep; this is not so different. We attribute the slightly increased sleep duration in the baseline controls (blue) to leaky Chrimson expression in ambient light. It is true that nighttime sleep in these strains does not replicate our Figure 5 results. We don’t know why, but our sleep induction protocol is during the day, so this does not alter our conclusions. It is possible that some long-term homeostatic processes are affected here in these particular strains, which this study does not focus on. We are not sure what the reviewer means by GFP RNAi. The knockdown clearly works (Figure 5—figure supplement 1); any minor discrepancies the reviewer noticed are probably due to leaky or cumulative Chrimson effects. These do not affect the conclusions we draw from the red light activation conditions.

6) I am not convinced the order of events (first arousal threshold, then sleep) is as the authors suggest. The former can be assayed instantly whereas the latter requires some time to qualify as sleep.

We agree that this was poorly framed in the first paper. We have changed the language (e.g., in the Abstract) to frame this differently, without losing any of the impact of our work regarding a distinct role for the dFB here. The reverse order (sleep first, then increased arousal thresholds) obviously makes no sense, so it is either simultaneous or the order we propose. It would be hard to imagine, however, how the engagement of sleep functions would not be preceded by increased arousal thresholds, and why the mechanisms involved in reducing behavioral responsiveness upon sleep initiation would not also operate during the final moments of wakefulness. We now make this more of a discussion point and less of a fait-accompli.

7) Lots of control data is missing for the behavior experiments. For example, GAL4 and UAS control are needed in Figure 2 and Figure 7.

Our controls are appropriate here: the exact same strain with or without ATR. This is accepted for *Drosophila* optogenetics. This applies to both Figure 2 and Figure 7. Perhaps the referee was confused because we did not show the 3 traces in Figure 7C, which would have been really messy. We show all of the summarised control data in Figure 7D. In other parts of the paper we also show heterozygous genetic controls, and data are consistent. To have to show both genetic controls, both of them with and without ATR, with and without red light, is excessive and would set a strange precedent for future optogenetic work. There is a good reason optogenetics is very appealing: these experiments are blessed with *internal* controls. For Chrimson, this works very well.

*8) Error bars are needed in Figure 2—figure supplement 2A and Figure 2—figure supplement 2E. And why is the behavioral response in Figure 2—figure supplement 2G on the baseline day is so different compared to Figure 2—figure supplement 2C on the baseline day? They should be the same genotype and condition. And what is the explanation why for the 10Hz light pulse activation of R23E10 has* no *effect on sleep duration?*

We apologise for the oversight and have corrected the figure (now Figure 3—figure supplement 3A). We agree that the 10Hz responsiveness data was a little messy and potentially confusing, so we took it out. Our goal for doing the 1Hz 5ms experiments was to provide a more physiological stimulation regime. Since we found similar results, that is sufficient. What other frequencies might or might not be doing is potentially interesting, but beyond the scope of this paper. In future work, we will be looking at a sweep of frequency effects.

Reviewer #3:[…]1) Figure 2 is essential in making the case that dFB activation reduces behavioral responsiveness separately from (and perhaps prior to) sleep. However, it is perplexing that responses are "abolished" (subsection section “Optogenetic activation of the sleep switch decreases behavioral responsiveness”) in wake flies with dFB activation, while in sleeping flies, the response is only diminished. Is this a function of needing to normalize to wake activity (Figure 2E)? If so, is wake activity (mean speed) affected by dFB activation? How about movement in response to a strong stimulus? This raises the more general possibility that locomotor function is simply different with dFB activation, as opposed to an actual change in responsiveness to the stimulus at a sensory level. Can the authors address this?

Figure 2 is indeed essential in making the case that dFB activation reduces behavioral responsiveness separately from sleep. We have adjusted the text, added more analyses, and corrected some mistakes to make sure this is absolutely clear. First the mistake: the reviewer was correct in suspecting that responsiveness is not ‘abolished’ in awake flies; rather, it is impaired. In our previous figure, we showed summary stats that looked like responsiveness was abolished (old Figure 2E, right panel). This was a mistake based upon poorly fitted responsiveness curves, where we reported the peaks of fitted curves. Since the awake dFB-activated flies had poor response, this was a poorly fitted curve, leading to what looked like a negative response. We have now corrected this mistake and show the raw data instead (mean peak responsiveness), where it is clear that awake dFB-activated flies respond less than controls (new Figure 2E, right panel).

These data are supported by the acute activation experiment that follows (Figure 2F). This experiment showed that only 1min of red light activation is sufficient to impair responsiveness, even in awake flies. We now provide an additional supplemental figure to address some of the questions raised by the reviewer (Figure 2—figure supplement 2), by focusing only on walking animals. Here we show that (A) Walking speed was not significantly different in the minute immediately preceding the stimulus, for the dFB-activated flies compared to controls; (B), Responsiveness is not significantly different between the experimental groups when the red light is off; (C) and (D), When we only analyse flies that are actively walking, we see a significant decrease in responsiveness in dFB activated flies compared to controls. So, these are both groups of flies that are awake and walking, but only the dFB-activated group is impaired in its behavioral responsiveness to the vibration stimulus.

2) Also in Figure 2, I am confused that the earlier work (Figure 2C and D, excluding sleep intensity) do not distinguish between wake and sleeping flies. Why not only analyze sleeping flies until shifting focus to wake flies in Figure 2E? If analysis of wake flies requires normalizing the baseline activity (Figure 2E right panel), then shouldn't the authors need to account for that in analyzing mean peak responsiveness (Figure 2D, for example)?

Our sleep intensity metric (Figure 2D, right) is exactly what the reviewer is looking for: responsiveness only in sleeping flies. The methodology for this is explained in Figure 2—figure supplement 1. Mean peak responsiveness (Figure 2D, left) indeed combines data from both mobile and immobile flies, as an average. However, we look at only awake flies in panel E and F, and in the new Figure 2—figure supplement 1. We’ve pretty much covered all aspects of the data.

3) Figure 2F is particularly critical, and the authors set it up as if it will answer something about whether the change in responsiveness or sleep comes first (subsection “Optogenetic activation of the sleep switch decreases behavioral responsiveness” "even in awake flies"). However, it appears to again mix/combine sleeping and awake flies for analysis, which is a major confound. In other words, what if most of these flies are rapidly put to sleep with the 1 minute dFB activation prior to stimulus? Then the figure simply shows that inducing sleep with dFB changes responsiveness, which we already know (from this work, but also from Donlea et al. (2011) and Donlea, Pimentel and Miesenbock (2014) – it is important to note that these published works have already demonstrated a change in arousability with dFB activation, even if not with such detail as shown here).

Please see our response to similar points above. Most of these flies are *not* rapidly put to sleep within the 1min of dFB activation, which is why the average speed is the same as baseline. We now provide analysis on only awake, walking flies as well, and the results are almost identical (this is because most flies were walking in the 1min before the stimulus, even with dFB activated). That said, some flies do become immobile very rapidly, and some gradually, leading to increased sleep duration and intensity after 5min. We could dissect these dynamics a little more, but the key point here is that responsiveness can be abolished in awake flies that are still walking around. This dissociates the responsiveness functions of the dFB from its sleep-inducing functions, at least temporally. This is a very critical observation, as the reviewer has noted. We hope this is now clearer.

Please note that we have also analysed the results from Figure 7 in awake flies only and see similar effects. The reviewer may have missed this as these results are stated in the text (subsection “Optogenetic activation of the sleep switch decreases behavioral responsiveness”). We can provide a figure for that if requested.

We appreciate that others have demonstrated increased arousal thresholds in dFB activated flies. However, the detail whereby we elaborate on the question of behavioral responsiveness is a major point of our paper. The dFB sleep studies that have previously addressed responsiveness have used it only as a ‘sanity check’ for sleep. Without that, it would not have been convincing to call these neurons sleep promoting. What we report here is a broader role for these neurons, to control behavioral responsiveness more generally. Arousal thresholds are more than just a sanity check for sleep studies. This is a necessary and complementary research direction for understanding sleep processes, and how some of these might operate during wakefulness as well. This is a major point of our paper, and we hope that this is clearer now.

4) The authors make a point of 1Hz and 10Hz optogenetic activation of dFB both changing responsiveness, but only 1Hz inducing sleep, therefore dissociating the two. However, these data are not convincing. In Figure 2—figure supplement 2G, why is the response so poor at baseline compared to C in this same figure? And why suddenly so much noise? Moreover, the change in responsiveness (H) appears similar +/- ATR: the baselines appear different and the 10Hz light causes a similar decrease in both (same issue in D of this figure). In other words, I am not convinced that there is a change in responsiveness, and therefore the ability to dissociate sleep time from arousal is hindered. The sleep intensity data is more convincing, but this is a comparatively less precise measure.

We disagree with the last point: sleep intensity is the more precise measure. With sleep intensity, we are looking at how only sleeping (immobile for >5min) flies respond to the mechanical stimulus, rather than an amalgam of awake and sleeping flies. We agree on the first point though, the 10 Hz data in panel G was messy. We’ve taken the 10Hz data out. It isn’t crucial for this study. We’ll redo it and put it use it somewhere else. It is interesting though.

5) In Figure 7, there are concerns. First, in contrast to Figure 5A, there is no longer an effect of INX6 RNAi expressed using R23E10-GAL4 (Figure 7A, left). Is there dilution of UAS-driven expression by co-expressing two UAS? Is INX6 knockdown even effective here? If we assume so, and if INX6 is mostly important in the context of dFB activation, then the authors should test these same measures following sleep deprivation. That would be most convincing with regard to a role for INX6 in modulating dFB driven sleep intensity. Overall, it would add a lot to this work for the authors to perform many of the experiments in Figure 2 after sleep deprivation, which is more physiologically relevant and should induce changes in behavioral responsiveness.

Please see our reply to reviewer #2 above, who had a similar question about these data. The suggestion to do sleep deprivation experiments is interesting: one prediction would be that less dFB activation would be required to achieve a similar level of behavioral inhibition. How to titrate that exactly would be an entirely new study. Rather, our study recurrently returns to electrophysiology to provide a level of understanding on the likely mechanisms involved, which we suggest pertains to oscillations generated by these neurons (as in Yap et al., 2017). We are hesitant at this point to engage in an entirely new manipulation (sleep deprivation) which might then need to be applied to all of our readouts, including electrophysiology. This is an excellent suggestion, but for a different study. That said, we have shown previously (in Yap et al., 2017), that 12 hours of dFB activation essentially produces a sleep-deprived fly (Figure 6E in Yap et al., 2017), consistent with our view that this is activating a distinct (transitional) sleep stage. There are bound to be confounds that will inevitably require a full-blown paper to explore and explain, hence our hesitation to approach this thorny area at the revision stage for this paper. We will be fully exploring these issues (and the unique genomics underlying dFB-induced sleep) in another paper (in preparation).